Composition and Vertical Flux of Particulate Organic Matter to the Oxygen Minimum Zone
of the Central Baltic Sea: Impact of a sporadic North Sea Inflow
Carolina Cisternas-Novoa[1]*, Frédéric A.C. Le Moigne[1,2], Anja Engel[1].
*[1] GEOMAR, Helmholtz Centre for Ocean Research Kiel, Düsternbrooker Weg 20, D-24105*
*Kiel*
*[2] Present address:* "Mediterranean Institute of Oceanography, UM 110, Aix Marseille Univ.,
Université 6 de Toulon, CNRS, IRD, 13288, Marseille, France"
*Corresponding author:* Carolina Cisternas-Novoa, GEOMAR, Helmholtz Centre for Ocean
Research Kiel, Düsternbrooker Weg 20, D-24105 Kiel, Germany, +49 431 600-4146
ccisternas@geomar.de
Keywords: Baltic Sea, Oxygen minimum zone, POC, PN, POP, TEP, CSP, Sediment trap,
Export efficiency.

**Abstract**

Particle sinking is a major form to transport photosynthetically fixed carbon below the euphotic zone via the biological carbon pump (BCP). Oxygen ($O_2$) depletion may improve the efficiency of the BCP. However, the mechanisms by which $O_2$-deficiency can enhance particulate organic matter (POM) vertical fluxes are not well understood. Here, we investigate the composition and vertical fluxes of POM in two deep basins of the Baltic Sea (GB: Gotland basin and LD: Landsort Deep). The two basins showed different oxygen regimes resulting from the intrusion of oxygen-rich water from the North Sea that ventilated the water column below 140 m in GB, but not in LD. In June 2015, we deployed surface-tethered drifting sediment traps in oxic surface waters (GB: 40 and 60 m; LD: 40 and 55m), within the oxygen minimum zone (OMZ, GB: 110 m and LD: 110 and 180 m), and at recently oxygenated waters by the North Sea inflow in GB (180 m). The primary objective of this study was to test the hypothesis that the different $O_2$ conditions in the water column of GB and LD affected the composition and vertical flux of sinking particles, and caused differences in export efficiency between those two basins.

Composition and vertical flux of sinking particles were different in GB and LD. In GB, particulate organic carbon (POC) flux was 18% lower in the shallowest trap (40 m) than in the deepest sediment trap (at 180 m). Particulate nitrogen (PN) and Coomassie stainable particles (CSP) fluxes decreased with depth, while particulate organic phosphorus (POP), biogenic silicate (BSi), chlorophyll *a* (Chl *a*), and transparent exopolymeric particles (TEP) fluxes peaked within the core of the OMZ (110 m); this coincided with the presence of manganese oxide (MnOx)-like particles aggregated with organic matter. In LD, vertical fluxes of POC, PN, and CSP decreased by 28, 42 and 56% respectively, from the surface to deep waters. POP, BSi and TEP fluxes did not decrease continuously with depth, but they were higher at 110 m. Although we observe higher vertical flux of POP, BSi and TEP coinciding with abundant MnOx-like particles at 110 m in both basins, the peak in the vertical flux of POM and MnOx-like particles was much higher in GB than in LD. Sinking particles were remarkably enriched in BSi, indicating that diatoms were preferentially included in sinking aggregates and/or there was an inclusion of lithogenic Si

(scavenged into sinking particles) in our analysis. During this study, the POC transfer efficiency (POC flux at 180 m over 40 m) was higher in GB (115%) than in LD (69%) suggesting that under anoxic conditions a smaller portion of the POC exported below the euphotic zone was transferred to 180 m than under re-oxygenated conditions present in GB. In addition, the vertical fluxes of MnOx-like particles were two orders of magnitude higher in GB than at LD. Our results suggest that POM aggregate with MnOx-like particles formed after the inflow of oxygen-rich water into GB, the formation of those MnOx-OM rich particles may alter the composition and vertical flux of POM, potentially contributing to a higher transfer efficiency of POC in GB. This idea is consistent with observations of fresher and less degraded organic matter in deep waters of GB than LD.

**1. Introduction**

Particle sinking is the primary mechanism for transporting photosynthetically fixed carbon below the euphotic zone via the biological carbon pump (BCP) (Boyd and Trull, 2007; Turner, 2015). Previous studies suggested that the transfer of particulate organic carbon (POC) from the euphotic zone to the ocean interior is enhanced in oxygen minimum zones (OMZs) (Cavan et al., 2017; Devol and Hartnett, 2001; Engel et al., 2017; Keil et al., 2016; Van Mooy et al., 2002). Possible mechanisms explaining the higher POC transfer include: i) the reduction of aggregate fragmentation due to the lower zooplankton abundance within the OMZ (Cavan et al., 2017; Keil et al., 2016); ii) the potentially high contribution of refractory terrestrial organic matter (OM) to the POC flux (Keil et al., 2016; Van Mooy et al., 2002); iii) a decrease in heterotrophic microbial activity due to oxygen ($O_2$) limitation (Devol and Hartnett, 2001); iv) the preferential degradation of nitrogen-rich organic compounds (Kalvelage et al. 2013; Van Mooy et al. 2002, Engel et al. 2017), and v) changes in ballast materials that may alter the sinking velocity and protect OM from degradation (Armstrong et al., 2002). Currently, the study of POC vertical flux in OMZ's has been mostly focused on the tropical ocean (Cavan et al., 2017; Devol and Hartnett, 2001; Engel et al., 2017; Keil et al., 2016; Van Mooy et al., 2002); whereas, how low $O_2$ concentration would

affect the composition and fate of sinking OM, and the efficiency of the BPC in oxygen-deficient
zones of temperate-boreal regimes such as the Baltic deep basins had been less studied.
The semi-enclosed, brackish Baltic Sea is a unique environment with strong natural gradients of
salinity and temperature (Kullenberg and Jacobsen, 1981), primary productivity, nutrients
(Andersen et al., 2017), and $O_2$ concentrations (Carstensen et al., 2014a). New production,
defined as the fraction of the autotrophic production supported by allochthonous sources of
nitrogen (Dugdale and Goering, 1967) is considered equivalent to the particulate OM export
(Eppley and Peterson, 1979; Legendre and Gosselin, 1989) on appropriate timescales. In the
Baltic Sea, new production varies seasonally (Thomas and Schneider, 1999); with periods of high
new production during spring and summer, supported by the diatom-dominated spring bloom and
by diazotrophic cyanobacteria respectively (Wasmund and Uhlig, 2003). Based on sediment trap
data, collected at 140 m depth in the Gotland Basin (GB), Struck et al. (2004) reported that the
highest fluxes of POC occurred in fall, followed by summer and spring. Using $\delta^{15}N$, they showed
that during the summer, $N_2$ fixation by diazotrophic species is the primary source (~41%) of the
exported nitrogen and that the majority of the sedimentary particulate organic matter (POM) in
the central Baltic Sea is of pelagic origin.
OM export from the euphotic zone to the seafloor has a dual significance in the deep basins of the
Baltic Sea. On the one hand, it contributes to the long-term burial of POC, and consequently to
the removal and long-term storage of $CO_2$ from surface waters (Emeis et al., 2000; Leipe et al.,
2011), and on the other hand, it connects the pelagic and the benthic systems contributing to the
$O_2$ consumption and hence deoxygenation at depth. Environmental and anthropogenic changes
may alter the magnitude and composition of OM transferred from the surface to the seafloor in the
Baltic Sea (Tamelander et al. 2017). The reduction of nutrient inputs as targeted by the Baltic
Marine Environment Protection Commission (HELCOM) may reduce in OM downward flux and
limit the oxygen depletion at depth. However, since hypoxia occurred naturally in the Baltic Sea
due to physical processes, mitigating eutrophication will only decrease the spatial extent and
intensity of the O2 deficiency in the deep basins.
GB (248 m) and Landsort Deep (LD, 460 m) are the deepest basins of the Baltic Sea. They exhibit
permanent bottom-water hypoxia (Conley et al. 2002), caused by a combination of limited water
exchange with the North Sea through the Kattegat, strong vertical stratification, and high
production /remineralization of OM due to eutrophication (Carstensen et al., 2014b; Conley et al.,
2009). A permanent transition zone of about 2 to 10 m thickness separates the oxygenated surface
and the oxygen-deficient waters, with a pelagic redoxcline located approximately between 127
and 129 m in GB, and between 79 and 85 m in LD (Glockzin et al., 2014). From the1950s to
1970s, the hypoxic zones (<60 µM) in the Baltic Sea had expanded fourfold (Carstensen et al.
2014). Salt-water inflows from the North Sea are the primary mechanism renewing deep water in
the central Baltic Sea (Günter et al., 2008). A Major Baltic Inflow (MBI) occurred in 2014/2015
(Mohrholz et al. 2015); this event ventilated bottom waters for five months between February and
July 2015 (Holtermann et al., 2017). This MBI caused the intrusion of $O_2$ to deep hypoxic waters,
substantial temperature variability (Holtermann et al., 2017), displacement of remnant stagnant
water masses by new water that changed the chemistry of the water column (Myllykangas et al.,
2017), and high turbidities that may be associated with redox reactions products (Schmale et al.,
2016). At the time of sampling (June 2015), the MBI had reached GB but did not affect LD,
located further northwest. The oxygenated water inflow reached GB at the beginning of March
and created a secondary near-bottom redoxcline (Schmale et al., 2016); the bottom water anoxia
started to re-established in July 2015 (Dellwig et al., 2018). In LD, water properties did not
change due to the MBI, the sulfidic layer was maintained (hydrogen sulfide, $H_2S$ concentrations
of 20.7- 21.2 µM), and salinity varied between 10.6 and 10.9 (Holtermann et al., 2017).
Pelagic redoxclines are the suboxic transition between oxic and anoxic - even sulfidic- waters. A
steep redox gradient characterizes this transition zone were electron acceptors and their reduced
counterparts are vertically segregated, and biogeochemical transformations mediated by microbial
processes are actively occurring (Bonaglia et al., 2016; Brettar and Rheinheimer, 1991; Neretin et
al., 2003). For instance, iron (Fe) and manganese (Mn) undergo rapidly reversible transformations
at the redox interface. Mn is an essential electron donor and acceptor in redox processes occurring
at brackish, pelagic systems with anoxic conditions like the deep basins of the Baltic Sea. Redox
conditions control the biogeochemical transformations between dissolved $Mn^{2+}$ and insoluble
oxides and hydroxides of $Mn^{4+}$. Under anoxic conditions dissolved reduced Mn forms dominates,
while in the presence of $O_2$ the formation of particulate manganese oxides (MnOx) is favored.
The concentration of dissolved Mn may reach 0.3 μM in GB and a maximum value of about 3
μM in the LD (Dellwig et al., 2012). van Hulten et al. (2017) estimated an aggregation threshold
for manganese oxides of 25 pM, and suggested that a minimal concentration of dissolved Mn is
required for an efficient aggregation and removal of MnOx. Therefore, in GB and LD, the balance
between dissolve Mn and the formation of MnOx is controlled by the $O_2$ availability (*e.g.,* Neretin
et al., 2003). LD is characterized by a permanently stratified water column and sulfidic bottom
waters; these conditions favored the accumulation of high concentrations of dissolved Mn
(Dellwig et al., 2012).
In contrast, GB is periodically affected by lateral intrusions of $O_2$ and the oxygenation of deep
water as a result of MBI that occur every one to four years (Matthäus and Franck, 1992), favoring
the occurrence of MnOx containing particles. MnOx production may be microbially mediated
(Richardson et al., 1988), or authigenic (Glockzin et al., 2014). In sulfidic waters, the reduction of
MnOx with sulfide occurs within a scale of seconds to minutes (Neretin et al., 2003), and is
inhibited by nitrate (Dollhopf et al., 2000). The oxygenation of the deep water of GB by the
2014/2015 MBI combined with the release of Mn from the sediments into the water column (Lenz
et al., 2015) generate appropriate conditions to enhance particulate MnOx formation and
vertically expand the zone where they could be observed in the water column.
MnOx-containing particles have previously been observed at pelagic redoxclines in the Baltic Sea
(Glockzin et al., 2014; Neretin et al., 2003). They are amorphous or star-shaped particles, and
occur as single particles or form aggregates with OM (Neretin et al., 2003), specifically with
transparent exopolymer particles (TEP) (Glockzin et al., 2014). The sinking velocity (0.76 m d$^{-1}$)
of those mixed aggregates containing MnOx and TEP was lower than what was predicted by
Stokes law possibly due to their star-shaped morphology and the high OM content. TEP are

highly sticky, polysaccharide-rich particles that can enhance particle aggregation rates and the formation of marine snow (Engel, 2000; Logan et al., 1995). Thus, the sinking of MnOx-OM aggregates may contribute to the downward flux of POC. However, high content of TEP relative to more dense particles could reduce the density of marine aggregates and decrease their sinking velocity (Engel and Schartau, 1999). Another type of less studied exopolymer particles are Coomassie stainable particles (CSP), they are protein-containing particles that stain with Coomassie brilliant blue (Long and Azam 1996). Little is known about the characteristics and dynamics of those particles in marine systems and their potential to form aggregates with MnOx had not been studied. Different to TEP, CSP have a limited role on the aggregation of diatoms (Prieto et al., 2002; Cisternas-Novoa et al., 2015), but seem to be important for the aggregation of cyanobacteria (Cisternas-Novoa et al., 2015). Mixed MnOx-OM aggregates may affect the cycling of particle-reactive elements like phosphorus and trace metals via scavenging processes, and it has been proposed that they could act as carriers of bacteria in the redoxcline (Dellwig et al., 2010). To date, there are no measurements of the density of MnOx-OM aggregates, their potential ballast effect on sinking OM, or their biogeochemical role modifying the vertical flux of POM in the Baltic Sea.

The objectives of this study are, first, to determine the amount and composition of particles sinking out of the euphotic zone into the deep basins of the Baltic Sea: GB and LD. Second, to study how the oxygenation of deep waters (>140 m) caused by the 2014/2015 MBI may affect the vertical flux of sinking particles. We, therefore, compared GB affected by the MBI with LD that was not affected and exhibited low $O_2$ concentration (>74 m) and even sulfidic conditions (>180 m). We hypothesized that the MBI that altered the water column chemistry and created different $O_2$ conditions in GB compared with LD affected the composition and vertical flux of sinking particles. Additionally, the higher abundance and *in-situ* formation of MnOx-OM aggregates may cause differences in degradation and export of OM between those two basins.

## 2. Methods

### 2.1. Sampling location and water column properties

Samples were collected during the BalticOM cruise in the Baltic Sea onboard the *RV Alkor* form
June 3[th] to June 19[th], 2015. We collected sinking particles using surface-tethered drifting sediment
traps (Engel et al., 2017; Knauer et al., 1979) in GB and LD (Table 1). Additionally, water
column samples (table 2) were collected using a Niskin-bottle rosette at the locations of the trap
deployments. Temperature, salinity and $O_2$ concentration were determined at each station using a
Sea-Bird (CTD) probe equipped with a $O_2$ sensor (Oxyguard, PreSens), calibrated with discrete
samples measured using the Winkler method (Strickland and Parsons, 1968; Wilhelm, 1888).
*2.2. Sediment trap design and deployment*
We deployed two surface-tethered drifting sediment traps for two days in GB, and one day in LD
(Fig.1). Each trap collected particles at four depths: 40 m (two arrays were deployed to evaluate
replicability of particle collection), 60 m (55m in LD), 110 m and 180 m (Table 1) to estimate
POM fluxes to and within the OMZ. 40 m was considered as the base of the euphotic zone based
on PAR measurements conducted during the cruise (data not shown). At each depth, 12 acrylic
particle interceptor tubes (PITs) mounted in a PVC cross frame were deployed. Each PIT was
equipped with an acrylic baffle at the top to minimize the collection of swimmers (Engel et al.,
2017; Knauer et al., 1979). The PITs were 7 cm in diameter and 53 cm in height with an aspect
ratio of 7.5 and a collection area of 0.0038 $m^{-2}$. The cross frame and PITs were attached to a line
that had a bottom weight and a set of surface and subsurface floats. The procedures for PIT
preparation and sample recovery followed Engel et al. (2017). Shortly before deployment, each
PIT was filled with 1.5 L of seawater previously filtered through a 0.2 μm pore size cartridge. A
preservative solution of saline brine (50 g $L^{-1}$) was added slowly to each PIT underneath the 1.5 L
of filtered seawater, carefully keeping the density gradient. The PITs were kept covered until
deployment and immediately after recovery to avoid contamination. After recovery, the density
gradient was visually verified, and the supernatant seawater was siphoned off the PIT. Then, we
pooled together the remaining water, containing the sinking material (~0.6-0.8 L), of 12 tubes per
depth into a large container, that we filled-up to 10 L with filtered seawater (between 0.4 and 1.5
L) to have the same volume per depth. After that, the samples were screened with a 500 μm mesh

to remove swimmers (Conte et al., 2001). Subsequently, samples were split into aliquots that were processed for the different biogeochemical analysis as described in Engel et al. (2017).

*2.3. Biogeochemical analysis*

Nutrients were measured in seawater samples collected in the deployment stations. Ammonium (detection limit of 0.05 μM) was measured directly on unfiltered seawater samples on board after Solórzano (1969). Phosphate, nitrate, and nitrite (detection limit of 0.04 μM) were filtered through a 0.2 μm pore size and stored frozen until their analysis; samples were measured photometrically with continuous flow analysis on an auto-analyzer (QuAAtro; Seal Analytical) after Grasshoff et al. (1999).

Particulate organic carbon (POC), nitrogen (PN), organic phosphorus (POP), and chlorophyll *a* (Chl *a*) were determined as described in Engel et al. (2017). Aliquots, of 100 to 200 mL of the trapped material and 500 mL of the seawater samples, were filtered in duplicate for each parameter at low vacuum (<200 mbar), onto pre-combusted GF/F filters (8h at 500°C). The filters were stored frozen (-20°C) until analysis. Prior analysis, filters for POC-PN determination were exposed to acid fumes (37% hydrochloric acid) to remove carbonates and subsequently dried for 12h at 60 °C. POC and PN concentrations were determined using an elemental analyzer (Euro EA, Hechatech) after Sharp (1974).

POP was analyzed after Hansen and Koroleff (1999). POP was oxidized to orthophosphate by heating the filters in 40 mL of deionized water (18.2MΩ) with Oxisolv (MERCK 112936) for 30 min in a pressure cooker. Orthophosphate was determined spectrophotometrically at 882 nm in a Shimadzu UV-VIS Spectrophotometer UV1201.

Chl *a* was analyzed after extraction with 10 mL of 90% acetone, the fluorescence of the samples was measured using a Turner fluorometer (440/685 nm, Turner, 10-AU) according to Strickland et al. (1972). The fluorometer was calibrated with a standard solution of Chl *a* (Sigma-Aldrich C-5753).

Biogenic silica (BSi) was determined in aliquots of 50 to 100 mL, filtered in duplicate onto 0.4 μm cellulose acetate filters. Samples were stored at -20°C until analysis. For the measurements,

filters were digested in NaOH at 85°C for 135 min; the pH was adjusted to 8 with HCl. Silicate
was measured spectrophotometrically according to Hansen and Koroleff (2007).
Polysaccharide (TEP) and protein (CSP) exopolymer particles, from sediment trap and water
column samples were analyzed by microscopy according to Engel (2009). Duplicate aliquots of 5
to 20 mL were filtered onto 0.4 μm Nuclepore membrane filters (Whatmann) and stained with 1
mL of Alcian Blue solution, a dye that target acidic polysaccharides, for TEP or 1 mL of
Coomassie brilliant blue solution, a dye commonly used to stain proteins (Bradford, 1976), for
CSP. Filters were transferred onto Cytoclear ® slides and frozen (-20°C) until microscopy
analysis. For the analysis, thirty images for each filter were captured under 200x magnification
using a light microscope (Zeiss Axio Scope A.1) connected to a color camera (AxioCam MRc).
Particle abundance and area were measured semi-automatically using an image analysis system
including the WCIF ImageJ software. The RGB was split into three channels: red, blue and green,
and the red was used to quantify the amount of TEP and CSP. Additionally, TEP and CSP in
water samples from the stations where we deployed sediment traps were analyzed
spectrophotometrically (with higher vertical resolution than microscopy) according to Passow and
Alldredge (1995) and Cisternas-Novoa et al. (2014), respectively. Concentrations of TEP are
reported relative to a xanthan gum standard and expressed in micrograms of xanthan gum
equivalents per liter (μg XG eq. L$^{-1}$), and concentrations of CSP are reported relative to a bovine
serum albumin standard and expressed in micrograms of bovine serum albumin equivalents per
liter (μg BSA eq. L$^{-1}$).
MnOx-containing particles have been commonly identified based on their morphology, size and
elemental composition, confirmed by scanning electron microscopy (SEM) and energy dispersive
x-ray microanalysis (EDX) (Neretin et al., 2003; Glockzin et al., 2014; Dellwig et al., 2010,
2018). In this study, we did not measure the elemental composition of the particles. Thus, we
identified them as "MnOx-like particles" based on similar morphology, size, and association with
organic matter (OM) as MnOx-containing particles previously described in the Baltic Sea (eg.,
Neretin et al., 2003 and Glockzin et al., 2014). The abundance and size of MnOx-like particles
were determined using particle recognition on filters and imaging processing similar to the
method used by Neretin et al. (2003) but without the chemical composition analysis of the
particles. For the image analysis, we used the same images as for TEP and CSP analysis and
modified image analysis procedure described above as follows: thirty images per filter (200x)
were analyzed semi-automatically using ImageJ software. After RGB split, the blue channel
pictures were used to quantify MnOx-like particles in the water column and sediment traps. In this
manner, the MnOx-like particles were clearly visible with a negligible disruption from TEP or
CSP stained blue.
Total hydrolyzable amino acids (TAA) were analyzed in unfiltered seawater and trapped material.
Samples were stored at -20°C until analysis. Duplicate samples were hydrolyzed at 100 °C in 6N
HCl (Suprapur® Hydrochloric acid 30%) and 11 mM ascorbic acid for 20h. Amino acids were
separated and measured by high-performance liquid chromatography (HPLC), after derivatization
with ortho-phthaldialdehyde using a fluorescence detector (Excitation/Emission 330/445 nm)
(Dittmar et al., 2009; Lindroth and Mopper, 1979). TAA concentrations were reported as µM of
monomer. The quantitative degradation index (DI) of Dauwe et al. (1999), based on changes in
amino acids composition of POM as it undergoes degradation processes, was calculated using the
factor coefficient of Dauwe et al. (1999) and the average and standard deviation of the TAA of
this data set.
Total combined carbohydrates (TCHO) > 1 kDa were determined by HPAEC-PAD according to
Engel and Händel (2011). TCHO were analyzed in the unfiltered seawater and sediment trap
material. Samples were stored at -20°C until analysis. Prior to analysis, the samples were desalted
by membrane dialysis using dialysis tubes with 1 kDa molecular weight cut-off (Spectra Por).
Desalination was conducted for 4.5h at 1°C. Then, a 2 mL subsample was sealed with 1.6 mL of
1M HCl in pre-combusted glass ampoules and hydrolyzed for 20h at 100°C. After hydrolysis, the
subsamples were neutralized by acid evaporation under $N_2$ atmosphere at 50°C, resuspended with
ultrapure Milli-Q water and analyzed on a Dionex 3000 ion chromatography system. TCHO
concentrations were reported as µM of monomer.
*2.4 Phytoplankton abundance*
Phytoplankton composition and abundance at the stations where we deployed sediment traps were
evaluated using light microscopy and flow cytometry. Counts of phytoplankton cells > 5 μm,
were made from 50 mL of fixed samples (Lugol's solution, 1% final concentration). Samples were
concentrated using gravitational settling and counted under a Zeiss Axiovert inverted microscope
(200x magnification) following the guidelines for determination of phytoplankton species
composition, abundance (HELCOM, 2012). The counts were made on either half (cyanobacteria,
diatoms, and *Dinophysis sp.*) or two strips (chryptophyta, unidentified dinoflagellates, and
chlorophyta) of the chamber. Individual filaments of cyanobacteria were counted in 50 μm length
units. The size of the counted phytoplankton species ranged from 10 to 200 μm.
Phytoplankton, <20 μm, cell abundance was quantified using a flow cytometer (FACSCalibur,
Becton, Dickson, Oxford, UK). 2 mL samples were fixed with formaldehyde (1% final
concentration) and stored frozen (-80 °C) until analysis (two weeks later). Red and orange
autofluorescence were used to identify chlorophyll and phycoerythrin cells. Cell counts were
determined with CellQuest software (Becton Dickenson); pico- and nanoplankton populations of
naturally containing chlorophyll and/or phycoerythrin (*i.e.*, Synechococcus) were identified and
enumerated.
*2.5 Statistics*
Significant differences between two parameters were tested using the Mann-Whitney U-test. The
results of statistical analyses were assumed to be significant at *p*-values < 0.05. Statistical
analyses were performed using Matlab software (MatlabR2014a).
**3. Results**
*3.1. Biogeochemistry of the water column*
At both stations, GB and LD, the water column was stratified during the study. In GB, the
seasonal thermocline was located between 22 and 37 m, with temperature decreasing rapidly from
9.8°C in the surface mix layer to 4.7°C below 37 m (Fig. 2a). Deeper in the water column, a
pycnocline (halocline) coincided with the oxycline and was located between 65 m (S=7.6) and 80
m (S=10.2); below 80 m the salinity gradually increased up to 13.5 (220 m). A hypoxic layer (<40
µM $O_2$) was located between 74 and 140 m; the core of the OMZ (<10 µM $O_2$) was located
between 96 and 125 m. The $O_2$ concentration increased from 35 µM $O_2$ at 140 m to 79 µM $O_2$ at
220 m (Fig. 2a). In LD, the seasonal thermocline was located between 10 and 39 m, where the
temperature decreased gradually from 12°C to 4.0°C (Fig. 2b). The pycnocline was between 55
(S=7.2) and 75 m (S=9) below that the salinity was constant (S=10.7) until the bottom of the
station (430 m). The $O_2$ concentration was below the detection limit (<3 µM $O_2$) from 74 m to the
deepest point sampled in LD (430 m).
The vertical profile of nutrients was different at both stations (Fig. 2). In GB, nitrate concentration
increased from below the detection limit in the upper ten meters to 0.17 µM at 40 m (Fig. 2a).
Concentrations were variable within the OMZ with 6 µM in the upper (80 m) and lower oxycline
(140 m), and 0.12 µM in the core of the OMZ (110 m). Nitrate concentration was 4.8 µM in the
deepest sample (220 m). Nitrite was below the detection limit in most of the water column except
for 60 m (0.09 µM) and 110 m (0.11 µM). Ammonium increased from 0.14 µM in the upper ten
meters to 1.15 µM at 40 m; concentrations were variable within the OMZ with less than 0.15 µM
in the upper (80 m) and lower oxycline (140 m), and maximum concentration of 3.28 µM in the
core of the OMZ (110m). Vertical profiles of phosphate and silicate at GB were similar; the
concentrations steadily increased from the upper ten meters of the water column (0.29 µM and
10.36 µM, respectively) to the OMZ (2.67 µM and 39.07 µM, respectively), and gradually
decreased below the OMZ (Fig. 2a). $H_2S$ was not detectable in GB.
In LD, nitrate and nitrite concentrations were below the detection limit between the surface and
250 m (<0.04 µM) (Fig. 2b). Nitrite showed a maximum of 0.22 µM at 350 m, and nitrate a
maximum of 6.0 µM at 400 m. Ammonium concentrations varied between 0.06 and 0.59 µM in
the upper 70 m and increased to 5.97 and 8.03 µM in the OMZ (>74 m). The lowest ammonium
concentration (0.07µM) was measured in the surface and the highest (8.03 µM) at 110 m.
Phosphate and silicate concentrations were relatively low within the mixed layer; gradually
increased below the pycnocline, and decreased again between 110 and 180 m. Phosphate
concentrations varied between 1.5 and 2.5 µM in the upper 110 m of the water column, decreased
to 0.22 μM at 180 m and increased to 2.7 μM at 430 m (deepest sample). Silicate ranged between
25 and 38 μM in the upper 110 m of the water column, decreased to 7.4 μM at 180 m, and
increased to 38.9 μM at 430 m. $H_2S$ was detectable below 180 m, with the highest concentration
(3.97 μM) at 250 m and the lowest (0.04 μM) between 300 and 350 m (Fig. 2b).
*3.2. Particulate organic matter concentration in the water column*
Chl *a* concentration in the upper 10 m was slightly higher in GB (1.5-1.7 μg L$^{-1}$, Fig. 3b) than in
LD (1.4-1.2 μg L$^{-1}$, Fig. 3e). At both stations, more than 90% of the total smaller phytoplankton
(<20 μm, pico- and nanophytoplankton) abundance, determined by flow cytometry, were
measured in the upper 60 m, although phytoplankton was detectable in the entire water column.
Pico- and nanophytoplankton abundance were 10% higher in GB than in LD (Table 2).
Picocyanobacteria determined by phycoerythrin fluorescence accounted for 92% and 96% of the
total picophytoplankton abundance in GB and LD, respectively. Picocyanobacteria abundance
was 30% higher in GB than in LD.
Phytoplankton (>5 μm) abundance, determined by microscopy, was 63% higher in LD than in GB
(Table 3). Filamentous cyanobacteria dominated the phytoplankton community at both stations
with up to 90% corresponding to *Aphanizomenon* sp. Cyanobacteria represented 56% of the
phytoplankton counts in GB and up to 74% in LD. Dinoflagellates (including mixotrophs),
dominated by *Dinophysis* sp, were significant in both stations (19% of the phytoplankton counts),
whereas chlorophytes (dominated by filaments of *Planctonema* sp. containing cylindrical cells)
were more abundant in GB than in LD (25% and 4% of the phytoplankton counts respectively).
Diatoms represented less than 1% of the phytoplankton in both stations, and they were slightly
more abundant at 40 m in LD (Table 3). BSi was higher in the upper 10 m (0.4-0.5 μM) and
decreased with depth in GB (Fig. 3b), whereas in LD, BSi showed a peak at 40 m and then
decreased with depth (Fig. 3e).
Vertical profiles of POC, PN, and POP concentration were similar in the water column of the two
stations (Fig. 3a, d). In GB, the concentrations were higher in the upper 10 m of the water column
(POC: 40.38 ± 0.80, PN: 3.89± 0.01, and POP: 0.26± 0.04 μM ) and decreased gradually with
depth until 110 m where relatively high concentrations (POC 18 ± 0.63, PN: 2± 0.08, and POP:
0.2 μM) were observed. The lowest concentrations were found at 180 m (POC: 11.97 ± 1.03, PN:
1.05± 0.02, and POP <0.03 μM) (Fig. 3a). In LD, POM decreased with depth from the surface
(POC: 35 ± 0.99, PN: 4± 0.09, and POP: 0.2 μM) to 40 m, remained relatively constant between
40 and 80 m and decreased again between 110 and 250 m (Fig. 3d).
We observed high concentrations of TEP and CSP in the upper 10 m in both stations. The highest
TEP concentration was determined at 1 and 10 m at both stations, and it was slightly higher (19%)
in GB than in LD (Fig. 3c, f). TEP and CSP vertical profiles were different from each other in GB
(Fig. 3c) and covaried in LD (Fig. 3f). Like observed for POC, PN, and POP, TEP concentrations
showed a peak at 110 m (50.29± 6.17 μg XG eq. $L^{-1}$) in GB. The highest concentration of CSP at
this station was observed in the shallowest (1 m) sample, CSP concentration decreased quickly
below 10 m, and then it increased at 140 and 220 m (the deepest sample, approximately 28 m
above the seafloor) (Fig. 3c). In LD, the highest concentrations of TEP and CSP were measured at
the surface (1 and 10 m) and at 110 m (Fig. 3f). TEP and CSP decreased with depth in the first 80
m (from 53.26± 7.10 to 18.39± 4.57 μg XG eq. $L^{-1}$ and from 53.26± 7.10 to 31.57± 18.78 μg BSA
eq. $L^{-1}$). Both types of gel-like particles showed an increase in concentration at 110 m (49.25±
4.08 μg XG eq. $L^{-1}$ and 66.89± 22.33 μg BSA eq. $L^{-1}$ respectively). Below 110 m, TEP
concentrations stayed relatively constant, while CSP concentrations decreased at 180 m and kept
relatively constant below that depth.
*3.3. MnOx-like particles vertical distribution in the water column*
Dark, star-shaped MnOx-like particles (Glockzin et al., 2014; Neretin et al., 2003) were only
observed below the fully oxygenated mixed layer in GB and, in less abundance, in LD (Fig. 4). In
GB, MnOx-like particles were observed from 80 m to 220 m; they appear as single particles and
forming large aggregates containing several MnOx-like particles associated with OM. Relatively
high concentration of MnOx-like particles ($2x10^6$ particles $L^{-1}$) were observed in the upper (80 m,
25 μM $O_2$) and lower (140 m, 36 μM $O_2$) oxycline, and at 220 m, 79 μM $O_2$ ($4x10^6$ particles $L^{-}$
$^1$)(Fig. 4a). The lowest abundance of MnOx-like particles ($7x10^5$ particles $L^{-1}$) was observed at

110 m, 6 μM $O_2$, i.e. in the core of the OMZ. The equivalent spherical diameter (ESD) of MnOx-

like particles varied between 0.6 and 30.5 μm, with a median size of 3.0 μm. The largest

aggregates (up to 30.5 μm) were observed in the upper oxycline (80 m). In LD, MnOx-like

particles were less abundant, smaller, and had a narrow distribution in the water column than in

GB. MnOx-like particles were not detected in the fully oxic (0-40 m) or fully anoxic (180 to 430

m) water column. At 60 m (135 μM $O_2$), right above the oxycline, MnOx-like particles began to

appear, however, in relatively low abundance. The maximum abundance of MnOx-like particles,

$9\times10^5$ $L^{-1}$, was observed in the oxycline at 70 m (27 μM $O_2$, Fig. 4b). The ESD ranged between

0.6 and 13.4 μm, the largest aggregates were observed at 70 m.

*3.4. Vertical flux of Sinking Particles*

Vertical fluxes of POC and PN varied little with depth in GB (Fig. 5a). POC flux slightly

increased by 18% from the shallowest (40 m) to the deepest (180 m) sediment trap. Fluxes of PN

(Fig. 5a) and CSP (Fig. 6b) were higher at 40 and 60 m and decreased (19 and 70 %) from 60 to

180 m respectively. On the other hand, fluxes of POP, BSi, Chl *a* (Fig. 5b) and TEP (Fig. 6a)

peaked in the sediment trap located in the core of the OMZ (110 m). The increment of fluxes at

110 m coincided with the high abundance of MnOx-like particles associated with TEP (Fig. 6a).

In addition, TEP size distribution, determined by image analysis, indicated an increase in large

TEP at 110 m (data not shown). In contrast, in LD, POC, PN (Fig. 5c) and CSP (Fig. 6d) fluxes,

steadily decreased with depth by 28, 42 and 56% from 40 to 180 m. Similar to the fluxes

measured in GB, the POP, BSi (Fig. 5d) and TEP (Fig. 6c) showed a smaller peak in the sediment

trap located at 110 m.

MnOx-like particles were drastically less abundant in sediment trap samples from LD than in GB,

and when present, they appeared as single particles, not aggregated with TEP or CSP (Fig. 6c, d).

At both stations, and similar to the water column samples, MnOx-like particles were not observed

in sediment trap samples collected in fully oxygenated waters (40 and 60 m). The flux of MnOx-

like particles at 110 and 180 m was two orders of magnitude larger in GB than in LD (Table 4). In

GB, MnOx-like particles occurred as single particles as well as aggregates with each other and

OM such as TEP and CSP (Figure 6a,b, and e), phytoplankton cells, or detrital material. The ESD of MnOx-like particles and aggregates collected in the traps ranged from 0.6 to 167 μm (median 2.8 μm) at 110 m and from 0.6 to 153 μm (median 3.3 μm) at 180 m. In LD, only a few, single MnOx-like particles were observed at 110 m (Fig. 6 c, d), their size ranged from 0.6 to 16.5 mm (median 1.8) (Table 4).

TAA flux ranged from 371±12 to 501± 33 μmol m$^{-2}$d$^{-1}$ in GB and from 502± 84 to 785± 54 μmol m$^{-2}$d$^{-1}$ in LD (Fig. 7a). In GB, the flux steadily decreased from surface to depth, whereas in LD the TAA flux at 40 m was lower than at 60 m and decreased with depth from 60 to 180 m (Fig. 7b). The vertical profile of TCHO flux was similar in both stations, although the magnitude of the flux was higher at LD. The TCHO flux varied between 303± 8 and 428± 14 μmol m$^{-2}$d$^{-1}$ in GB (Fig. 7a) and between 503± 19 and 584± 8 μmol m$^{-2}$d$^{-1}$ in LD (Fig. 7b). At both stations, TCHO fluxes increased from 40 to 110 m, where the highest flux was measured, and then it decreased at 180 m.

3.5. *Chemical composition of sinking and suspended particles*

Comparing molar elemental ratios of sinking (from sediment trap material) and suspended (from water column) particles to the revisited Redfield ratio for living plankton (106C: 16N: 15Si: P; Redfield et al., 1963; Brzezinski, 1985), our results showed that the POC:PN ratio of sinking particles was slightly above this ratio at both stations. The POC:PN ratios of sinking particles in GB and LD were not significantly different. In GB however, ratios increased with depth from 9.8 to 12.6, while in LD it varied between 11.1 and 15.4 without a clear trend with deep. The POC:POP ratio of sinking particles was lower ($p<0.05$; Mann–Whitney U-test) in GB (90.1-244) than in LD (230-772) with the highest value observed at 40 m and the lowest at 110 m. At both stations the POC:BSi ratios varied between 1.7 and 4.2 and PN:BSi ratios varied between 0.2 and 0.4; the lowest values were observed at 110 m (Table 5).

Contrastingly, in suspended particles, POC:PN ratios were higher in GB than in LD ($p<0.001$). In GB, it varied between 8.4 and 12 without a clear trend with depth; while in LD, it decreased with depth from 8.7 (at 1m) to 6.2 (at 400 m), and a slightly higher value of 7.8 was observed at 430 m.

The POC:PN and POC:POP were significantly higher (p<0.01) in sinking than in suspended
particles (Table 5). The POC:BSi and the PN:BSi ratios were much lower in sinking than in
suspended particles at both stations (GB: $p<0.05$; LD: $p<0.01$). In sinking particles, the POC:BSi
ratio was below Redfield ratio of 7, whereas it was one to two orders of magnitude higher in
suspended particles (Table 5). The PN:POP ratio was significantly lower in sinking ( 0.15-0.43)
than in suspended particles (9.7-44.5) at both stations ($p<0.001$). In sinking particles, it was
always below the Redfield ratio of 16, while in suspended particles, it was in the range of
Redfield ratio in the upper 80 m in GB and always above in LD.
At both stations, the contribution of AA to POC was more significant in sinking than in
suspended particles. Similarly, the carbon contained in TCHO made up a larger percentage in
sinking than in suspended particles (Table 5). The amino acid-based degradation index (DI,
Dauwe et al., 1999) varied from 0.1 to 1.14 in sinking OM and was higher than in suspended OM
(-1.25 to -0.42) in both stations. In sinking OM, the DI decreased with depth in GB, whereas in
LD, there was not a clear trend with depth (Table 5). The DI was higher in GB than in LD in
sinking as well as in suspended OM.
**4.  Discussion**
In this study, we 1) characterized the biogeochemistry of the water column and the sinking
particles in GB and LD, during early summer 2015, and 2) determined the vertical flux of sinking
particles in those two deep basins of the Baltic Sea. Our results suggested that the intrusion of
oxygenated water to GB, as consequence of the 2014/2015 MBI, caused changes in the water
chemistry that affected the chemical composition and degradation stage of the sinking and
suspended particles. Consequently, the composition and magnitude of the sinking particle flux
were different in GB and LD.
*4.1 Physical and biogeochemical conditions in GB and LD*
In general, physical and biogeochemical conditions (temperature, salinity, $O_2$, and inorganic
nutrient concentrations) were similar in the euphotic zone of both stations. Moreover, though
there were slight differences between the stations concerning phytoplankton abundance and
composition, and concentration and chemical composition of POM, in the surface water column,
those were not significant. The concentration of Chl *a* (Fig. 3) and the abundance of pico- and
nano-phytoplankton (Table 2) were slightly higher (20 and 10 % respectively) in GB than in LD.
This agrees with estimates of integrated total primary production (PP), which were 10% higher in
GB (380 mg C $m^{-2}$ $d^{-1}$) than in LD (334 mg C $m^{-2}$ $d^{-1}$; Piontek et al., unpublished). At both
stations, the abundance of pico-phytoplankton (<2 μm) was an order of magnitude higher than
nano-plankton (Table 2). These findings coincided with what was described previously for early
summer in the Baltic Sea that indicate that during this period the productivity is sustained mostly
by pico- and nano-phytoplankton communities (Leppänen et al., 1995) which co-existed with
cyanobacteria and other phytoplankton species (Kreus et al. 2015). Microscopic analysis, on the
other hand, indicated that phytoplankton (>5 μm) abundance was 47% higher in LD than in GB.
At both stations, filamentous cyanobacteria (> 90% *Aphanizomenon* sp.) were numerically the
predominant type (55 and 74% of the phytoplankton counts in GB and LD respectively),
dinoflagellates (including mixotrophs) correspond to 20%, and diatoms correspond to >1% of the
phytoplankton abundance in the upper 40 m (Table 3). Diatoms were slightly higher in LD than in
GB, and this coincide with a small peak in BSi concentration (1.5 μM, Fig. 3e) at 40 m in LD.
Although at both stations the diatoms proportion from the total phytoplankton abundance was
negligible, they could make a difference in the composition of sinking particles leaving the
euphotic zone in LD due to selective aggregation of diatoms (Passow et al., 1991); however, in
both stations sinking particles showed a similar enrichment in BSi. The low abundance of diatoms
relative to cyanobacteria in the euphotic zone indicated that at both stations, the spring bloom was
terminated and cyanobacteria were starting to build up the summer bloom that generally occurs in
June-July (Kreus et al., 2015); *Aphanizomenon* sp. and *Nodularia spumigena*, are known to form
summer blooms, where they accumulate at the sea surface of the thermally stratified water
column (Bianchi et al., 2000; Nausch et al., 2009; Wasmund, 1997).
The concentration of particulate elements (POC, PN, POP, BSi) was slightly higher in the surface
waters of GB compared to LD; while polysaccharide (TEP) and protein (CSP) containing
exopolymeric particles were in similar abundance at both stations. TEP and CSP were more
abundant in the euphotic zone, which supports the idea of a phytoplankton origin; however, the
concentration of TEP in this study was 69% (in GB) and 76% (in LD) lower than previously
reported for summer in the central Baltic Sea (Engel et al., 2002). Likewise, our dissolved
inorganic nitrogen concentrations were below the detection limit in the surface, while phosphate
concentrations were higher (>0.3 μM) than observed in the Engel et al. (2002) study. Mari and
Burd (1998) reported that TEP concentration peaked during the spring bloom and in summer in
the Kattegat. TEP production may be enhanced by environmental conditions such as nutrient
limitation (Mari et al., 2005; Passow, 2002), which are characteristic of late summer in the Baltic
Sea (Mari and Burd 1998). In the Baltic Sea, the spring bloom (March-April) is usually followed
by a period of reduced PP (Chl-$a$ ~ 2 μgL$^{-1}$) that preceded the cyanobacteria summer bloom,
typically observed in June-July (Kreus et at., 2015). Surface satellite-derived Chl-$a$ concentrations
(MODIS) in GB indicate a constant increment from mid-May to mid-June2015 (Le Moigne et al.,
2017); our monthly Chl-$a$ concentrations derived from VIIRS for June 2015 in the Baltic Sea
(Fig.1) showed similar Chl-$a$ concentrations. Considering this trend in Chl-$a$ concentration and
the availability of phosphate in the water column, we could assume that our samples were
collected at the beginning of the summer bloom (middle June). In general ecosystem models from
the Baltic Sea indicate that the termination of the summer bloom depends upon the phosphate
availability (Kreus et at., 2015). Thus, likely TEP concentrations had not reached the higher value
previously observed after summer bloom when inorganic nutrients were depleted. Although
satellite-derived Chl-$a$ concentrations is a valuable tool to evaluate the trend of the PP, the
magnitude of the concentration of Chl-$a$ from remote sensing is difficult to estimate in the Baltic
Sea (Darecki and Stramski, 2004). The concentration of Chl-$a$ in GB and LD derived from direct
measurements were much lower (~1.5 μg L$^{-1}$), suggesting that our samples were collected during
a period of low phytoplankton biomass typically observed before the summer bloom. In any case,
the concentration of phosphate was not limiting the system. Another possible explanation for the
rather low concentrations of TEP could be their removal from the surface by aggregation and

subsequent sedimentation during the spring bloom due to the high abundance of cells and detrital particles during this time (Engel et al., 2002) and the relatively low grazing pressure that lead to higher export after the spring bloom (Lignell et al., 1993).

Although the composition and amount of OM in the surface waters at the two trap stations were similar, below the euphotic zone (40 m) the vertical profile of nutrients and particulate matter concentrations were distinctly different; likely due to the 2014/2015 MBI (Holtermann et al., 2017) that reached the deep waters of GB. This inflow replaced the old stagnant water masses by new water masses (Schmale et al., 2016), changing the salinity in the deepest waters and the vertical distribution of $O_2$ increasing its concentrations below140 m and constraining the oxygen-deficient layers from 74 to 140 m depth. The combination of physical effects (the displacement of water masses, turbulent mixing and lateral transport) and the consequent development of redox conditions through 2015 may have impacted the distribution of MnOx-like particles and POM in GB. In addition to changes in $O_2$ concentration, the MBI altered the redox conditions in GB creating a secondary redoxcline at 140 m, where concentrations of $O_2$ and MnOx-like particles increased. One consequence of those changes is the vertical extension of the layer in which MnOx-containing aggregates could form (Schmale et al., 2016); a previous study showed that MnOx might precipitate from the water column of GB following a MBI event (Lenz et al., 2015). POC and PN concentrations peaked at 110 m, this higher concentration at 110 m was even more evident in POP and TEP, while CSP concentration peaked at 140 m (Fig. 3); this is the first study that examines the potential role of CSP in forming aggregates with MnOx-containing particles. The highest concentration of MnO-like particles (Fig. 4a) in the water column was not observed at 110 m (the core of the OMZ) but at 80 m (oxycline), and below 140 m in the newly oxygenated water layers.

In contrast, LD maintained permanent suboxic (<5 μM $O_2$) waters below 74 m as $H_2S$ was detectable below 180 m. Below 100 m the vertical profiles of POM and BSi did not change with depth. The only exception was TEP and CSP concentration that similar to in GB peaked at 110 m and MnOx- like particles showed a small increment at 70 m (in the oxycline). This suggest that,

similar to the results of Glockzin et al (2014), the MnOx-like particles, abundant in the oxycline
may form sinking aggregates with TEP and CSP, then, when those aggregates sunk to anoxic
waters (below 74 m), the MnOx-like particles may dissolve releasing TEP and CSP to the water
column, where CSP concentration decreased quickly likely due to microbial degradation, but the
concentration of TEP remain constant to the bottom of LD.
MBI can have a significant impact on nutrient recycling. In GB nitrate concentration increased
possibly as a consequence of the oxidation of reduced nitrogen compounds (e.g., ammonium,
ammonia and organic nitrogen compounds like urea) (Le Moigne et al., 2017) that accumulated
during the stagnation (anoxic) period previous to the MBI (Hannig et al., 2007). Scavenging of
phosphate onto Mn or Fe oxides has been shown in previous studies (Neretin et al., 2003).
Phosphate can bind to Fe hydroxides and MnOx and settle down during oxic conditions, building
up a phosphate pool in the sediments that later on when the $O_2$ decreases may become a source of
phosphate (Gustafsson and Stigebrandt, 2007). Moreover, Myllykangas et al. (2017) reported that
the new water masses intruded during 2014/2015 MBI displaced the stagnant water masses in GB.
Thus, the low concentrations of silicate and phosphate that we measured in the deep waters of GB
may also be a direct consequence of the intrusion of oxygenated, low-nutrient waters associated
with the MBI. In contrast, in LD, the water column remained anoxic down to the sea floor (430
m), below the oxycline an increase of ammonium was observed (Fig.2b), which could be an
indicator for anaerobic respiration of OM, e.g., denitrification (Bonaglia et al., 2016; Hietanen et
al., 2012).
In summary, though GB and LD had similar surface conditions in terms of phytoplankton
production and POM stocks, during this study, we found differences in the vertical concentration
of nutrients (Fig. 2) and POM (Fig. 3) between GB, ventilated by the MBI, and LD, a station that
remained suboxic. Our results suggest that the MBI caused differences in the vertical profile of $O_2$
that modified the redox conditions of the water column and enhance the *in-situ* formation of
MnOx-like particles (Fig. 4). Alternatively, the inflow may transport new MnOx-like particles to
GB. Those abundant MnOx-like particles may aggregate with POM in GB, influencing the
vertical distribution of POM in the water column.
*4.2 Potential influence of $O_2$ concentration and redox conditions on vertical flux of sinking*
*particles in GB and LD*
During this study, we also investigated the effect of different $O_2$ concentrations and redox
conditions on the fluxes of particles. Our measurement of POC flux at 40 m, below the euphotic
zone, were $11.7 \pm 0.82$ mmol C m$^{-2}$ d$^{-1}$ in GB and $19.8 \pm 1.22$ mmol C m$^{-2}$ d$^{-1}$ in LD. Extrapolating
those measurements to annual flux, we obtain $4.37 \pm 0.31$ mol C m$^{-2}$ yr$^{-1}$ in GB and $7.44 \pm 0.46$ mol
C m$^{-2}$ yr$^{-1}$ in LD. Our results from GB are in the same range as the estimation derived from a
biogeochemical model; *i.e.* 3.8 - 4.2 mol C m$^{-2}$ yr$^{-1}$ (Kreus et al., 2015; Sandberg et al., 2000;
Stigebrandt, 1991) for the Baltic Sea; however, our results from LD are higher than the annual
POC fluxes predicted by those models. The high POC flux observed in this study is not surprising
since it represented one (in LD) and two (in GB) days in June when the POC vertical flux out of
the euphotic zone is relatively higher in the Baltic Sea compared with late fall and winter. The
biogeochemical model of Kreus et al. (personal communication)  estimated that POC flux in June
ranged between 8 and 13 mmol m-$^2$d-$^1$; this is in the same range that our observations.
One of the main advantages of our sediment traps is that we can study the flux of sinking particles
at various depths simultaneously (i.e. higher vertical resolution). Therefore, we measured the
POM flux in oxic waters (40 m and 60 (55) m); at the core of the OMZ (110 m) at 180 m in both
basins. Traps located a 180 m depth collected particles in sulfidic waters at LD and in recently
oxygenated waters (affected by the MBI) in GB. The vertical flux of POM and BSi was different
at the two studied basins; for example, POC flux was between 25 and 40% higher in the upper
110 m of the LD than in GB (even though the PP was 10% higher in GB). However, the POC
fluxes at 180 m (deepest trap) weresimilar in both basins; indicating a substantial decrease in the
POC flux between 110 and 180 m at the LD. The POC flux (and the PN flux which showed a
similar vertical profile) did not decrease with depth in the GB. In contrast, in the LD there was a
reduction of 17 and 16% of the POC flux from 40 m and 60 m (in the oxycline) and from 110 to
180 m respectively; the POC flux did not change from 60 to 110 m when a large section of the
water column was suboxic ($O_2 < 5$ μM from 74 m to the bottom of the station). From 110 to 180
m the water column was completely anoxic, and $H_2S$ was detectable at 180 m. The high flux of
POC at GB coincided with the appearance of dark, star-shaped particles that we defined as
MnOx-like particles, particularly evident at GB (Fig. 6a,b, and e), but also present in LD.  Based
on their morphology, size, and aggregation with OM, we propose that those particles correspond
to MnOx-containing particles enriched in OM that have been previously described at GB (Neretin
et al., 2003; Pohl et al., 2004; Glockzin et al., 2014; Dellwig et al., 2010, 2018) and LD (Glockzin
et al., 2014; Dellwig et al., 2010). The higher flux of MnOx-like particles in GB than in LD is
probably due to the oxygenation and change in the deep water redox conditions that enhance the
formation of MnOx-like particles associated with OM. This suggests that the reduction of the
POC flux below 110 m in the LD may be related to the $O_2$ depletion and the absence of MnOx-
OM aggregates in the anoxic zone.
The POP flux was similar in the oxic water column (up to 60 m) in both basins; however, it was
almost two and three times higher at 110 and 180 m respectively in GB than in LD. A peak in the
POP and BSi flux was observed at 110 m in both basins, but the magnitude of the increment was
much higher GB than in LD. In GB the POP flux increased 62% from 60 to 110 m (OMZ) and
then decreased by 28% from 110 to 180 m. Vertical flux of POP, BSi, and Chl-*a* (Fig. 5) were
enhanced at 110 m, which coincide with the high flux of MnOx-like particles. This high flux of
MnOx-like particles is maintained at 180 m, while the POP, BSi and Chl-*a* flux decreased at this
depth. This vertical distribution is is likely due to the enhanced formation of MnOx-like particles
in the hypoxic layer (<40 μM $O_2$) located between 74 and 140 m that may scavenge POP, and
aggregate with cells or phytodetritus containing BSi and Chl-*a*. Although the POP flux peaked at
110 m in LD as well, the increment was only 30 % from 60 m (suboxic) to 110 m (anoxic), and it
decreased by 78% from 110 to 180 m (sulfidic waters), these variations with depth were also
observed in the BSi flux. In LD, the flux and size of MnOx-like particles were much smaller than
in GB, and they were more abundant at 110 m than at 180 m.
Similar to the vertical distribution of POM in the water column discussed in section 4.1,
differences in POM and BSi fluxes between basins are likely associated with the large inflow of
oxygen-rich saltwater that displaced the old-stagnant water masses and changed the chemistry of
the water column (Myllykangas et al., 2017). Under euxinic conditions (*e.i.,* scenario observed in
LD without the influence of the MBI), the maximum concentration of particulate Mn is found in
the oxycline (Glockzin et al., 2014). Below the oxycline, and due to the presence of $H_2S$, the
particulate Mn concentration decreased drastically. During this study, we observed a high
concentration of MnOx-like particles flux at 110 and 180 m (Table 5) in GB, in agreement with
the high flux of particulate Mn measured in sediment traps located at 186 m in June 2015
(Dellwig et al., 2018). The oxygenation of the deep water layers of GB by the MBI caused the
absence of $H_2S$ (Schmale et al., 2016) and provided redox conditions favorable for the formation
of MnOx, resulting in the high MnOx-like particles flux measured in the sediment trap located in
the core of the OMZ (110 m) and at 180 m (oxygenated deep water). There were two possible
sources of MnOx associated with the 2014/2015 MBI in GB. On the one hand, the lateral
transport of low-density aggregates formed by MnOx and OM (Glockzin et al., 2014), and on the
other hand, the *in-situ* formation and deposition of MnOx following the oxygenation of the water
column (Dellwig et al., 2018). In clear contrast to the oxygenated deep layers of GB, in LD, we
measured $H_2S$ below 180 m, this could explain why although those aggregates were present in
this station at 110 m, they may dissolve in sulfidic waters, thus, were not as abundant and did not
form aggregates with TEP (Fig.6c).
The presence of MnOx-like particles in aggregates (Fig 6a) may have implications for the vertical
flux of POC, PN and POP in a stratified system with a pelagic redoxcline like the Baltic Sea.
Under steady state, the upward diffusion and oxidation rates of the dissolved Mn are balanced by
the sinking and dissolution rates of MnOx. During Mn-oxidation, the MnOx could aggregate with
POM and trace metals. Then, in the sulfidic waters, slow-sinking MnOx enriched in OM will be
dissolved liberating the OM and altering the vertical distribution and the flux of all associated
particle elements (Glockzin et al., 2014). This has been previously observed in other anoxic
basins; for example, in the Cariaco Basin, total particulate phosphorus reached their maximum
flux in sediment traps close to the redoxcline (Benitez-Nelson et al., 2004; Benitez-Nelson et al.,
2007). Moreover, even in the anoxic zone, the abundant aggregate associated bacteria (Grossart et
al., 2006) could partially or entirely degrade the organic compounds in those particles using $NO_3^-$
or MnOx as an electron acceptor. This may explain why we observed a clear peak in the vertical
fluxes of POP, BSi, Chl-*a* (Fig. 3a, b), TEP (Fig. 6a) and TCHO (Fig. 7a) at 110 m, followed by a
small decrease at 180 m in GB. In LD a smaller increment in the vertical fluxes of POP, BSi (Fig.
3d), TEP (Fig. 6c) and TCHO (Fig. 7b) were also observed. The vertical fluxes of those
compounds coincided with the abundance of MnOx-like particles; we assume that the MnOx
aggregated not only with TEP as described before (Glockzin et al. 2014) and observed in this
study (Fig. 6a), but also with aggregates containing phytoplankton cells and phytodetritus that
may enhance POP, BSi, Chl *a*, and TCHO export. On the other hand, nitrogen-rich components of
POM like PN (Fig. 3a), TAA (Fig. 7a), and CSP (Fig. 6a) gradually decreased with depth in GB,
suggesting that those compounds were less scavenge by MnOx-OM rich aggregates.
Primary production (PP) in GB was 10% higher than in LD during our study (Piontek et al.
unpublished data). However, the POC flux below the euphotic zone (at 40 m) was 42% higher in
LD than in GB and comparable at both stations at 180 m. The fraction of PP exported as POC is
termed export production (*e-ratio*) (Buesseler et al., 1992), and it is calculated as the POC flux
below the euphotic zone divided by the PP. We calculated the *e-ratio* using [14]C-based PP
measurements (Piontek et al. unpublished data) and carbon flux at 40 m (shallowest sediment trap
depth, considered at the base of the euphotic zone). The *e-ratio* was larger in LD (0.77) compared
to GB (0.41)*; i.e.*, the percentage of the PP exported as POC below the euphotic zone was 77% in
LD versus 41% in GB. This suggests that either a higher proportion of the PP was remineralized
in the euphotic zone of GB compared with LD, or particles were sinking faster in LD than in GB
likely due to differences in composition. On the other hand, the transfer efficiency of POC to the
deeper water column (*i.e.,* the ratio of POC flux at 180 m over POC flux at 40 m) was higher in
GB (115%) than in LD (69%). The transfer efficiency of POM is largely controlled by the
remineralization rate and the sinking velocity of particles (De La Rocha and Passow, 2007;
McDonnell et al., 2015; Trull et al., 2008). The higher POC transfer efficiency in GB than in LD
can be attributable to differences in the sinking velocities of the particles in those two stations.
Particulate MnOx may sink through the redoxcline in GB (Neretin et al., 2003) acting as ballast
material and nucleus for MnOx-OM rich aggregates formation. Those aggregates could have sunk
more quickly, limiting the time spent in the water column and the degradation by particle-
attached microbes. Assuming that MnOx-like particles had a density between 1.5 and 2.0 g cm$^{-3}$
(Glockzin et al., 2014), the largest particles measured at GB (167 μm, Table 4) will have a sinking
velocity based on Stokes' law between 508 and 1014 m d$^{-1}$. If we consider a mixed aggregate that
is 50% TEP, density 0.9 g cm$^{-3}$ (Azetsu-Scott and Passow, 2004) and 50% MnOx ( density 1.5 g
cm$^{-3}$), its density would be 1.2 g cm$^{-3}$, and its theoretical sinking velocity will be 204 m d$^{-1}$. This
indicates that, theoretically, the largest mixed aggregates composed of MnOx and TEP observed
in GB could reach 180 m (the location of our deepest sediment trap) in less than one day.
However, the average measured sinking velocity of MnOx-containing particles in the laboratory
for particles between 2 and 20 μm was 0.76 m d$^{-1}$, which is significantly lower than the theoretical
value (Glockzin et al., 2014). Glockzin et al. (2014) suggested that the star shape and the content
of OM were responsible for the lower than predicted sinking velocity. There is no information
about the amount of OM relatively to MnOx-containing particles in those mixed aggregates, or
how the MnOx to OM ratio may affect the density and sinking velocity of larger aggregates like
the ones we observed. Due to the shape and size of MnOx-OM aggregates observed in our study
(Fig. 6e), we could assume those are the same type of aggregates described before by Glockzin et
al. (2014). Although we did not measure the sinking velocity of those aggregates, we did observe
a higher abundance of them associated with TEP at 110 and 180 m in GB than in LD. Thus, the
formation of MnOx aggregates rich in OM could represent an additional mechanism (see
introduction) to explain why the efficiency of the OM export is different under anoxic than under
oxic conditions in the Baltic Sea. The oxygenation of anoxic deep water in GB caused by the
2014/2015 MBI, may have led to enhanced precipitation of manganese, iron, and phosphorus
particles (Dellwig et al., 2010; Dellwig et al., 2018). For example, the formation of P-rich, metal
oxides precipitates occur in the anoxic waters of the Black Sea (Shaffer, 1986) and Cariaco Basin
(Benitez-Nelson et al., 2004; Benitez-Nelson et al., 2007) were higher concentration of particulate
inorganic and organic phosphorus have been observed in sediment traps close to the redoxcline.
Alternatively, BSi could also act as ballast material incrementing the sinking velocity of marine
aggregates (Armstrong et al., 2001; Klaas and Archer, 2002). Our results showed that sinking
particles were strongly enriched in BSi relatively to C and N and compared to suspended particles
that were depleted in BSi (Table 5). Diatoms are the major  phytoplankton group that produces
BSi to build their cell walls (Martin-Jézéquel et al., 2000), and they are the dominant
phytoplankton species during the spring bloom. However, during our study, diatoms represented
less than 1% of the phytoplankton abundance in the water column, and even though there was a
strong enrichment in BSi in the sinking particles, this was similar in GB and LD (Table 5).
Therefore, either the differences in export production nor in transfer efficiency between GB and
LD could not be solely explained by the amount of diatoms cells, phytodetritus or BSi in sinking
particles at those two basins.
*4.3 Differences on composition and lability of sinking and suspended organic matter in GB and*

734        *LD*

In the sections above we compared the biogeochemical conditions and the size of the POM pool
in the euphotic zone of GB and LD. We then looked at how the sinking flux of OM was affected
by the different $O_2$ concentrations in the water column. Now, we focus on the influence of $O_2$ in
the chemical composition of sinking and suspended particles. Suspended or slow sinking particles
that spend more time in the water column should theoretically, show a more substantial degree of
degradation (Goutx et al., 2007). Relative to the Redfield molar ratio: 106 POC:16 PN:15 BSi:
POP. POM showed enrichment in POC relative to PN and POP, especially in sinking particles
from LD and suspended particles from GB. Our measured values of POC:PN (~10) and POC:POP
(between 89 and 506) in suspended OM coincide with the simulated ratio reported immediately
after the culmination of the spring bloom by Kreus et al. (2015). The same study had suggested
that POC:POP higher than Redfield ratio might lead to an enhancement of particle export (Kreus
et al., 2015), however, no direct observations had confirmed this hypothesis. Our measurements
showed that the relative higher POC:POP ratios in sinking OM from LD, compared with GB, do
not lead to a higher transfer efficiency at this station. Compared to the suspended OM in LD, the
POP content was lower in GB, possible related to scavenging of POP into MnOx aggregates (see
section 3.4).
In addition, at both stations, sinking particles were strongly enriched in BSi (Table 5) probably
due to the preferential sinking of diatoms and remnants diatom-rich detritus from the spring
bloom. Differently, suspended particles had a relatively low content of BSi; this is not surprising
considering the small proportions of diatoms in the euphotic zone at the time of our sampling. The
concentration of BSi decreased below the detection limit from 60 m in the GB, and 70 m in the
LD. This observation coincides with previous studies reporting selective incorporation of diatoms
into sinking aggregates in the Baltic Sea (Engel et al., 2002; Passow, 1991), whereas non-diatoms
species, although they may be abundant in the suspended phytoplankton, may not be present in
sinking particles (Passow, 1991).
Another explanation for of higher BSi content in sinking particles may be the inclusion of
lithogenic Si in our measurements; lithogenic Si may have been present in the water column or
being transported by laterally adverted material. A recent study suggests that contributions of non-
biogenic sources could be significant during alkaline extraction (Barão et al., 2015). The even
more substantial enrichment in BSi observed in sinking particles from 110 m in both basins, may
result from adsorption and/or co-precipitation of silica in sinking particles containing MnOx
(Dellwig et al., 2010; Hartmann, 1985); or by the formation of aggregates that are enriched in
MnOx as well as in phytodetritus from diatom origin.
The TAA based degradation index, DI (Dauwe et al. 1999) covers a wide range of alteration
stages; the more negative the DI, the more degraded the samples, positive DI indicates fresh
organic matter. In our study, the sediment trap material had a DI between 0.10 and 1.14, while
suspended OM has a DI between -0.26 and -1.25 (Table 5). These values coincide with what
reported earlier by Dauwe et al. (1999), and indicate that: first, the sinking particles collected in
the sediment traps were less altered (they have a more positive DI) than the suspended OM
collected in the Niskin bottle. Second, sinking particles from GB were fresher than the ones from
LD, and the degradation stage increased with depth in both stations. The higher contribution of
AA and CHO to the POC pool in sinking than in suspended OM and the AA- DI indicates that
suspended OM was more degraded than sinking OM. The highest degree of degradation in
suspended OM and sinking OM from LD may be the result of a long time that light suspended
OM or slow sinking particles spend exposed to degradation in fully oxygenated surface waters
than dense, fast sinking particles collected in sediment traps.
The higher abundance of aggregates, formed by a combination of MnOx-like particles and OM,
observed at 110 and 180 m in GB could act as bacteria hot spots that combined with a higher $O_2$
concentration in GB may increase the microbial degradation on sinking particles collected in GB.
However, the AA-DI, indicated that sinking OM was less altered, and therefore more labile than
the sinking OM in LD. This implied that in addition to the higher transfer efficiency of POC in
GB (see discussion above); the OM reaching the seafloor was fresher and less degraded. This
supports the idea that mix aggregates composed by MnOx and OM may be larger and faster
sinking than the previously described by Glockzin et al. (2014). This explanation is mostly
speculative, and based on the observation of large mixed aggregates in the 110 and 180 m traps
(Fig. 6, Table 4). However, as mention in the previous section, further work on directly
determining sinking velocity is required to prove this hypothesis.
**Conclusion**
Fluxes and composition of sinking particles were different in two deep basins in the Baltic Sea:
GB and LD during early summer 2015. The two stations had similar surface characteristics and
POM stock; however, at depth, the vertical profile of the POM concentration, as well as the
vertical flux of sinking particles was different, likely related to differences in the $O_2$
concentration. The 2014/2015 MBI supplied oxygen-rich waters to GB transporting solid material
from shallower areas and modifying the $O_2$ vertical profile and the redox conditions in the
otherwise permanent suboxic deep waters. This event did not affect LD allowing for the
comparison of POM fluxes and composition under two different $O_2$ concentrations with similar
surface water conditions. Export efficiency (*e-ratio*) derived from *in-situ* PP measurements and
POC flux derivate from sediment traps indicated higher export efficiency in LD than in GB.
However, the transfer efficiency (POC flux at 180 m over POC flux at 40 m) suggested that under
anoxic conditions found in LD, a smaller portion of the POC exported below the euphotic zone
was transferred to 180 m than under oxygenated conditions present in GB. The MBI also transport
solid Mn from shallower areas towards GB deep that may have contributed to the higher
abundance of MnOx-OM in GB. Our results suggest that a new possible mechanism to explain the
differences in the OM fluxes under different $O_2$ concentration could be the formation and
prevalence of aggregates composed of MnOx and organic matter in GB. Those aggregates were
significantly larger and more abundant in GB compared to LD where sulfidic waters constrained
their presence. Our results indicate that at GB not only a higher proportion of the POM leaving
the euphotic zone reached our deepest sediment trap, but also that this POM was fresher and less
degraded. We propose that after a MBI in GB, the aggregates containing MnOx-like particles and
organic matter could have reached the sediments relatively fast and unaltered, scavenging not
only phosphorus and TEP, as described previously, but also other compounds like BSi, POP and
CSP. The higher fraction of sinking particles exported below the euphotic zone and reaching 180
m in GB suggest that at this station a significant fraction of the POM could reach the sediments,
50 m below our deepest sediment trap, relatively unaltered. The remineralization of the organic
matter reaching the sediments may contribute to the quick re-establishment of anoxic conditions
in the sediment-water interface in GB. The relevance of this process needs to be further
investigated in order to be included in $O_2$ budget and long-term predictions of the MBI impact in
the $O_2$ and OM cycles.
**Author Contributions**
C.C.N. designed and performed the sediment trap work at sea, analyzed samples and wrote the
manuscript. F.A.C.L.M, designed and performed the sediment trap work at sea and contributed to
the writing of the manuscript. A.E designed and participated in the scientific program at sea and
discussed and commented on the manuscript.
**Acknowledgements**
This research was supported by the DFG Collaborative Research Center 754 "Climate-
Biogeochemistry Interactions in the Tropical Ocean" (to A.E., C.C.N. and F.A.C.L.M), by a
Fellowship of the Excellence Cluster 'The Future Ocean' (CP1403 to F.A.C.L.M.), and by a
DAAD short term grant (57130097 to C.C.N.). We thank Jon Roa, Tania Klüver, Scarlett Sett,
Angela Stippkugel, Carola Wagner, Clarissa Karthäuser, Moritz Ehrlich, Sonja Endres, Hannes
Wagner, Ruth Flerus, Sven Sturm and Christian Begler for support during traps preparation and
deployments, help with experiment or analyzed samples. We Thank Judith Piontek for her
contribution to the design of the scientific program at sea, Jaime Soto- Neira for useful discussion
and help with figure preparation and Cindy Lee for helpful advices.

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

**Figure Captions**

Figure 1. Monthly averaged Chl *a* distribution derived from VIIRS for June 2015 in the Baltic Sea. Black circle and "x" indicate the position of the trap deployment and the seawater collection, respectively, in Gotland Deep (GB) and Landsort Deep (LD). The lower panel shows the trajectory of the trap deployed at GB and LD.

Figure 2. Water column profiles at the location of the sediment trap deployments in (A) GB, and (B) LD. Left panel: oxygen (blue), temperature (red), and salinity (black). Middle panel: nitrate ($NO_3$, white squares), nitrite ($NO_2$, grey circles), and ammonium ($NH_4$, black triangles). Right panel: phosphate ($PO_4$, grey diamond), and silicate ($Si(OH)_4$, black circles). Grey lines indicate the depths at which we deployed sediment traps.

Figure 3. Vertical profiles of concentration of particulate organic carbon (POC), particulate nitrogen (PN), and particulate organic phosphorus (POP) in GB (A) and LD (D); vertical profiles of concentration of chlorophyll *a* (Chl *a*) and biogenic silicate (BSi) in GB (B) and LD (D); and vertical profiles of concentration of transparent exopolymeric particles (TEP) and Coomassie stainable particles (CSP) in GB (C) and LD (F) Grey lines as figure 2.

Figure 4. Vertical profiles of MnOx-like particles and $O_2$ concentration in the water column at the location of the sediment traps deployments. (A) GB and (B) LD. Grey lines as in figure 3.

Figure 5. Vertical fluxes of particulate organic carbon (POC) and particulate nitrogen (PN) as well as oxygen concentration in GB (A) and LD (C). Vertical fluxes of particulate organic phosphorus (POP), biogenic silica (BSi) and chlorophyll *a* (Chl *a*) in GB (B) and LD (D).

Figure 6. TEP and CSP fluxes in GB (A and B) and LD (C and D). In addition to vertical fluxes, each profile is complemented with microscopic images (200x) of material collected at each depth. In GB, star-shaped MnOx-like particles are clearly visible as single particles and forming aggregates with TEP (A), and CSP (B). MnOx-like particles were less abundant in LD (C and D). (F) A larger magnification (400x) image of MnOx-like particles at 110 m showing more detail on the shape of those particles and aggregates formed with TEP.

Figure 7. Vertical fluxes of total hydrolyzable amino acids (TAA) and total carbohydrates (TCHO) as well as oxygen concentration in (A) GB, and (B) LD.

Table 1. Sediment traps deployment and recovery locations, dates, collection times and depths.Two sediment traps were deployed at 40 m (A and B) to evaluate replicability.

| Station | Lat | Lon | Date | Station depth | Deployment time (d) | Trap depths (m) |
|---|---|---|---|---|---|---|
| Gotland Basin (GB) | 57.21 °N | 20.03 °E | 08/06/2015 | 248 m | 2 | 40A, 40B, 60, 110, and 180m |
| | 57.27 °N | 20.25 °E | 10/06/2015 | | | |
| Landsort Deep | 58.69 °N | 18.55 °E | 15/06/2015 | 460 m | 1 | 40A, 40B, 55, 110, and 180m |
| (LD) | 58.68 °N | 18.68 °E | 16/06/2015 | | | |

Table 2. Abundance of chlorophyll and phycoerythrin containing pico- and nano-plankton measured by flow cytometry in GB and LD.

| | Depth (m) | Phytoplankton (cells mL$^{-1}$) | | | Cyanobacteria-like (cells mL$^{-1}$) | | |
|---|---|---|---|---|---|---|---|
| | | picoplankton | nanoplankton | Total | picoplankton | nanoplankton | Total |
| GB | 1 | 87963 | 2097 | 90060 | 5225 | 731 | 5956 |
| | 10 | 94369 | 2628 | 96997 | 8795 | 920 | 9716 |
| | 40 | 4999 | 68 | 5067 | 2174 | 69 | 2243 |
| | 60 | 4125 | 35 | 4160 | 1990 | 42 | 2032 |
| | 80 | 599 | 7 | 606 | 238 | 15 | 253 |
| | 110 | 594 | 7 | 601 | 326 | 29 | 356 |
| | 140 | 1144 | 14 | 1158 | 356 | 2 | 358 |
| | 180 | 908 | 9 | 917 | 366 | 20 | 385 |
| | 220 | 2270 | 19 | 2289 | 1063 | 34 | 1097 |
| LD | 1 | 92359 | 2283 | 94642 | 834 | 177 | 1011 |
| | 10 | 86426 | 1708 | 88134 | 2990 | 232 | 3223 |
| | 40 | 2022 | 92 | 2114 | 2243 | 69 | 2312 |
| | 60 | 1524 | 62 | 1586 | 1294 | 24 | 1318 |
| | 70 | 908 | 43 | 951 | 613 | 17 | 630 |
| | 110 | 1735 | 82 | 1817 | 1181 | 17 | 1198 |
| | 180 | 1339 | 75 | 1415 | 946 | 34 | 980 |
| | 250 | 1593 | 82 | 1676 | 949 | 36 | 985 |
| | 300 | 1521 | 48 | 1569 | 1047 | 17 | 1064 |
| | 350 | 1608 | 57 | 1665 | 908 | 12 | 920 |
| | 400 | 1548 | 73 | 1621 | 1047 | 22 | 1069 |
| | 430 | 1562 | 68 | 1631 | 875 | 19 | 894 |

Table 3. Phytoplankton abundance analyzed microscopically for samples collected at the location of trap deployment in GB and LD.

| | | Phytoplankton Abundance (L$^{-1}$) | | | | | | | |
| | | GB | | | | LD | | | |
| | | 1 m | 10 m | 40 m | Total | 1 m | 10 m | 40 m | Total |
|---|---|---|---|---|---|---|---|---|---|
| Cyanobacteria* | Total | 14148 | 13536 | 0 | 27684 | 37368 | 32526 | 96 | 69990 |
| Chryptophyta | Total | 140 | 112 | 28 | 280 | 1400 | 882 | 56 | 2338 |
| Bacillariophyceae | Total | 96 | 94 | 44 | 234 | 462 | 112 | 102 | 676 |
| | *Chaetoceros* sp. | 58 | 42 | 24 | 124 | 434 | 106 | 26 | 566 |
| | *Skeletonema* sp. | 26 | 8 | 12 | 46 | 12 | 0 | 8 | 20 |
| | *Thalassiosira* sp. | 12 | 44 | 8 | 64 | 16 | 6 | 68 | 90 |
| Dinophyceae** | Total | 3772 | 4424 | 1192 | 9388 | 9032 | 7662 | 1404 | 18098 |
| | *Dinophysis sp.* | 678 | 742 | 2 | 1422 | 450 | 214 | 4 | 668 |
| | other | 3094 | 3682 | 1190 | 7966 | 8582 | 7448 | 1400 | 17430 |
| Chlorophyta | Total | 5320 | 6860 | 28 | 12208 | 2072 | 1022 | 238 | 3332 |
| | *Planctonema* sp. | 5320 | 6860 | 28 | 12208 | 2072 | 1022 | 238 | 3332 |

*Filamentous cyanobacteria were counted in 50 µm length units ( >90% were *Aphanizomenon* sp.)

**Include mixotrophs

Table 4. MnOx-like particle fluxes and size as equivalent spherical diameter (ESD) determined by image analysis in GB and LD.

| Station | Depth (m) | MnOx-like particles ($cm^2 m^{-2}d^{-1}$) | Median size ESD ($\mu m$) | Size range ESD ($\mu m$) |
|---------|-----------|-------------------------------------------|---------------------------|--------------------------|
| GB | 110 | 5666± 994 | 2.8 | 0.6-167 |
| | 180 | 7789± 955 | 3.3 | 0.6-153 |
| LD | 110 | 50.3±1.8 | 1.8 | 0.6-16.5 |
| | 180 | 2.6±0.3 | 1.4 | 1.2-9.3 |

Table 5. Amino acids (AA), carbohydrates (CHO), elemental molar ratios and amino acid-based degradation index of sinking and suspended particles in GB and in LD.

| | | Depth (m) | AA-C:POC % | CHO-C:POC % | POC:PN | POC:POP | POC:BSi | PN:BSi | PN:POP | DI |
|---|---|---|---|---|---|---|---|---|---|---|
| Sinking particles | GB | 40 | 19.2 | 18.3 | 9.8 | 244 | 3.9 | 0.4 | 24.9 | 1.49 |
| | | 40 | 17.6 | 17.2 | 9.4 | 222 | 4.1 | 0.4 | 23.6 | 1.43 |
| | | 60 | 15.8 | 17.6 | 9.5 | 232 | 2.8 | 0.3 | 24.3 | 1.13 |
| | | 110 | 13.9 | 22.2 | 11.3 | 90.1 | 1.7 | 0.2 | 8.0 | 0.71 |
| | | 180 | 11.1 | 18.5 | 12.7 | 123 | 3.0 | 0.2 | 9.7 | -0.03 |
| | LD | 40 | 13.5 | 9.4 | 12.2 | 772 | 3.6 | 0.3 | 63.4 | 0.30 |
| | | 40 | 14.3 | 8.4 | 11.1 | 413 | 4.1 | 0.4 | 37.2 | 0.27 |
| | | 55 | 19.1 | 11.0 | 12.4 | 332 | 3.0 | 0.2 | 26.7 | -0.02 |
| | | 110 | 13.4 | 12.0 | 15.4 | 230 | 2.7 | 0.2 | 14.9 | 0.11 |
| | | 180 | 14.3 | 12.9 | 15.3 | 341 | 4.2 | 0.3 | 22.3 | -0.29 |
| Suspended particles | GB | 1 | 8.2 | 16.9 | 10.4 | 155 | 91.4 | 8.8 | 14.9 | |
| | | 10 | 10.8 | 8.8 | 10.5 | 151 | 87.1 | 8.3 | 14.4 | |
| | | 40 | 4.9 | 2.8 | 9.2 | 88.8 | 134 | 15 | 9.7 | -0.81 |
| | | 60 | 5.4 | 2.7 | 9.8 | 127 | 125 | 13 | 13.0 | -0.27 |
| | | 80 | 4.7 | 0.00 | 10.4 | 145 | | | 13.9 | |
| | | 110 | 9.0 | 6.6 | 8.5 | 245 | | | 29.0 | 0.98 |
| | | 140 | 5.3 | 0.00 | 10.6 | 283 | | | 26.7 | |
| | | 180 | 5.7 | 4.3 | 11.4 | 506 | | | 44.5 | -0.40 |
| | | 220 | 8.6 | 3.3 | 12.1 | 271 | | | 22.5 | |
| | LD | 1 | 7.0 | 0.00 | 8.7 | 205 | 515 | 59.5 | 23.7 | |
| | | 10 | 13.0 | 9.1 | 8.4 | 196 | 101 | 12.0 | 23.3 | |
| | | 40 | 0.00 | 8.9 | 8.1 | 336 | 24.5 | 3.0 | 41.5 | -0.53 |
| | | 60 | 6.1 | 10.3 | 7.8 | 301 | 16.9 | 2.2 | 38.4 | -0.12 |
| | | 70 | 7.9 | 10.7 | 7.7 | 292 | 248 | 32.1 | 37.9 | |
| | | 110 | 12.2 | 5.4 | 7.9 | 225 | | | 28.3 | 0.80 |

| | | | | | | |
|------|------|------|-----|-----|------|------|
| 180  | 10.1 | 11.3 | 7.0 | 205 | 29.2 | 0.34 |
| 250  | 12.0 | 8.8  | 6.5 | 249 | 38.2 |      |
| 300  | 10.9 | 0.00 | 6.7 | 137 | 20.4 |      |
| 350  | 10.7 | 10.1 | 6.8 | 146 | 21.6 |      |
| 400  | 10.0 | 0.00 | 6.2 | 230 | 37.2 |      |
| 430  | 9.4  | 9.5  | 7.8 | 149 | 19.0 |      |

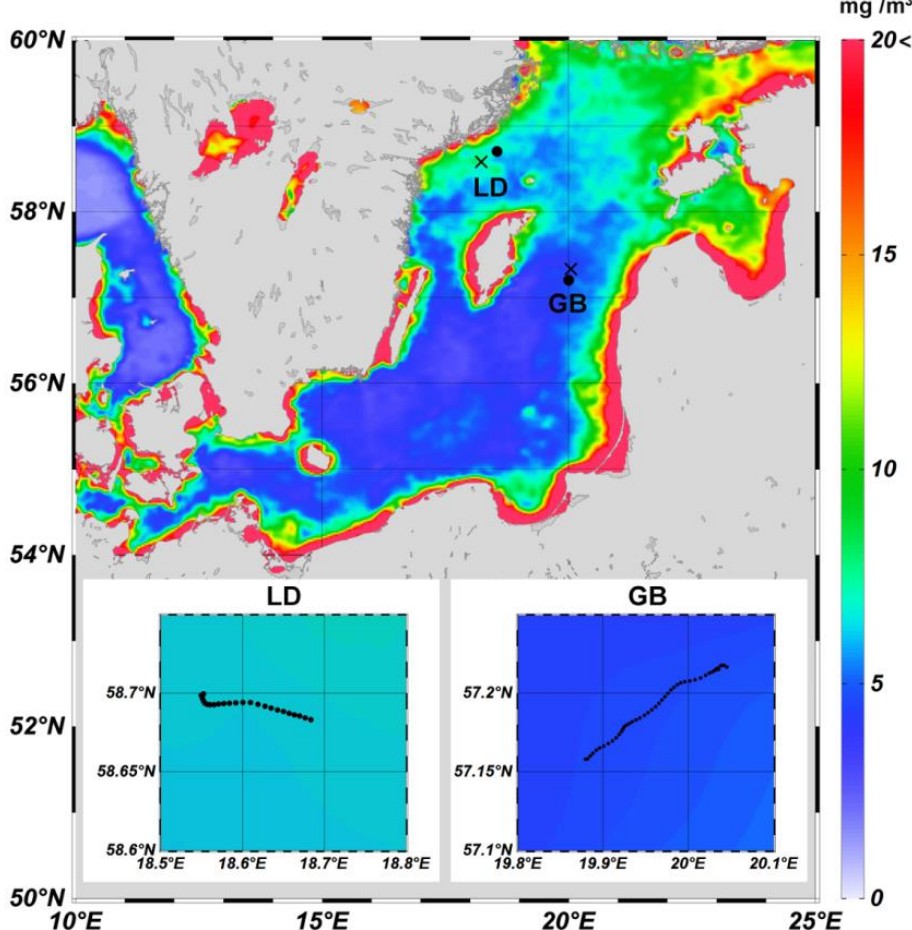

Fig. 1

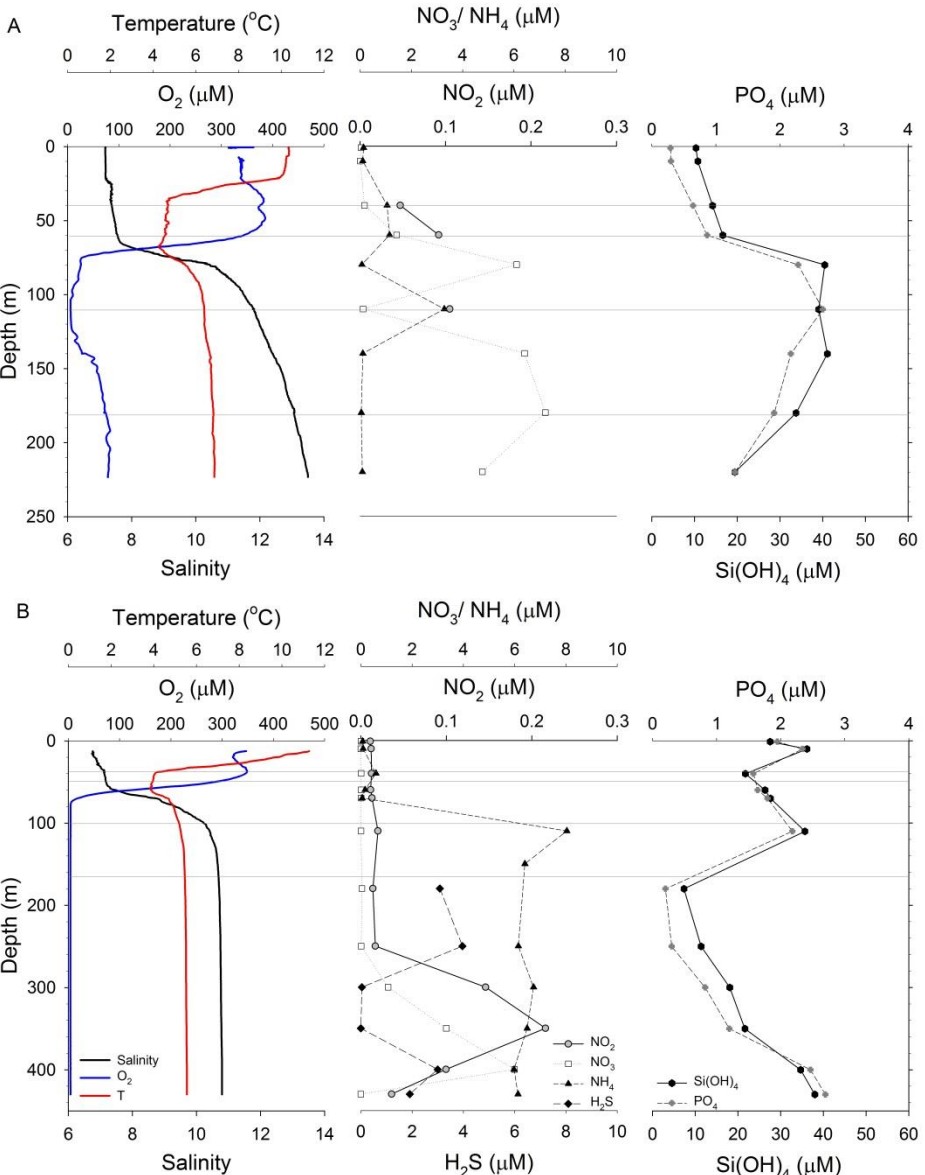

Fig. 2

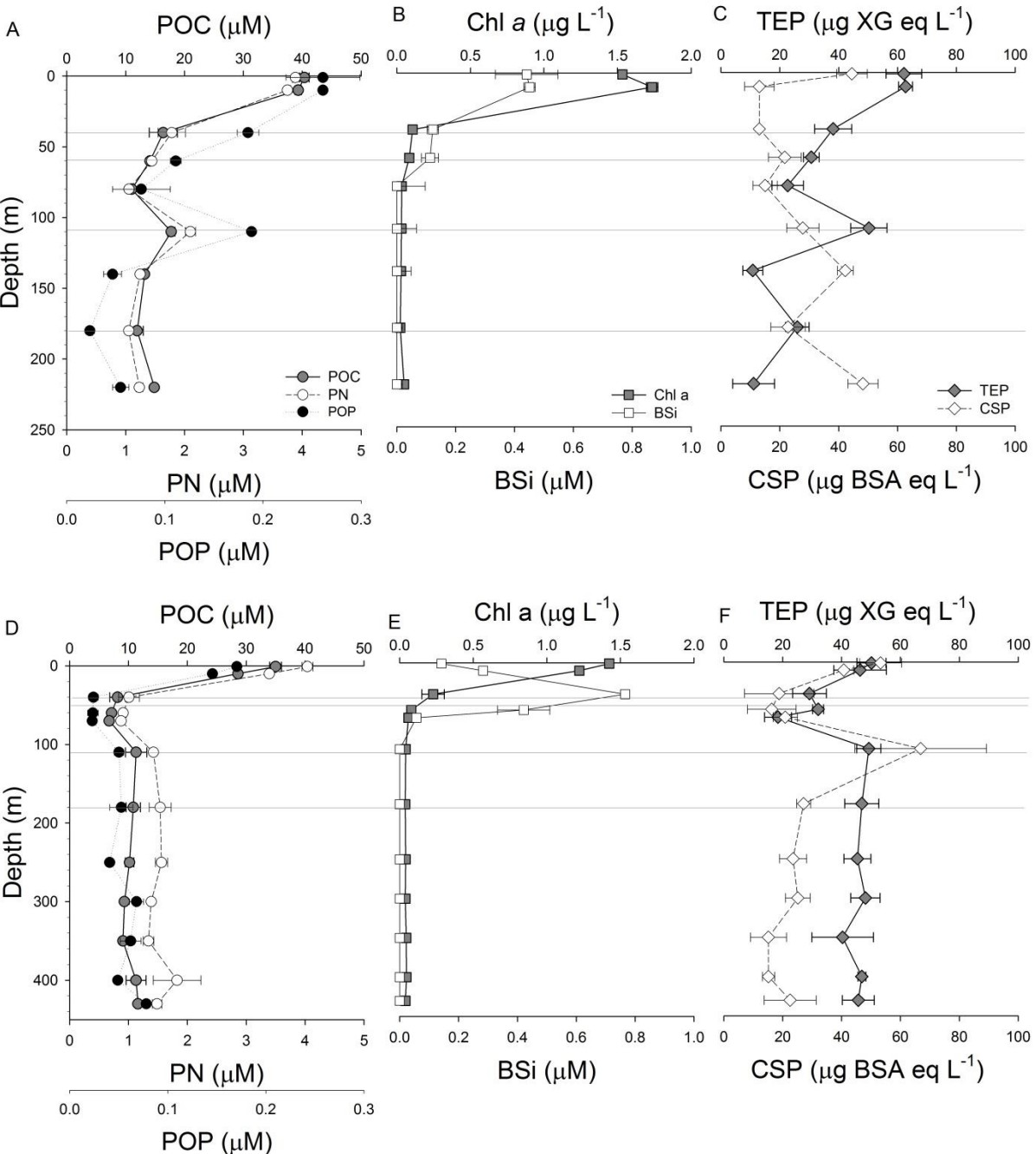

Fig. 3

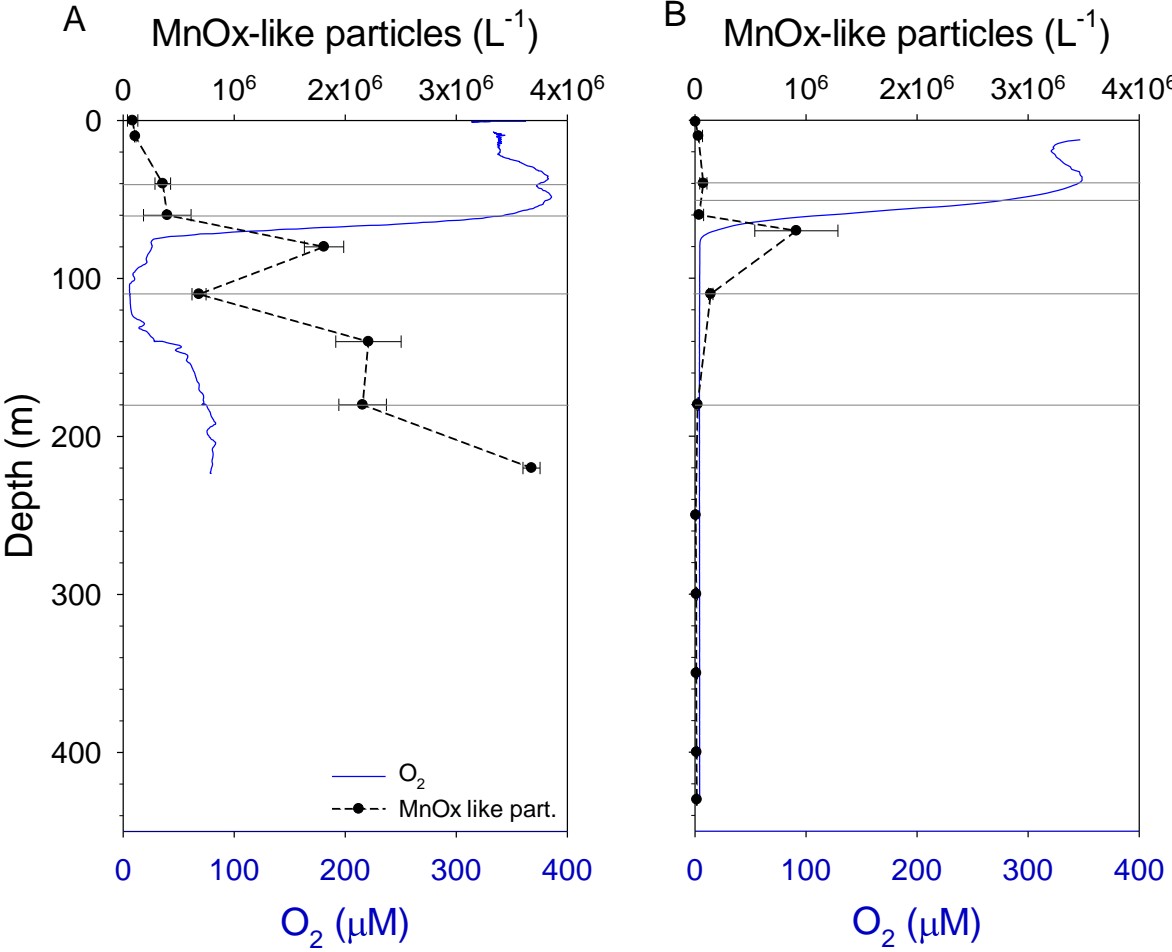

Fig. 4

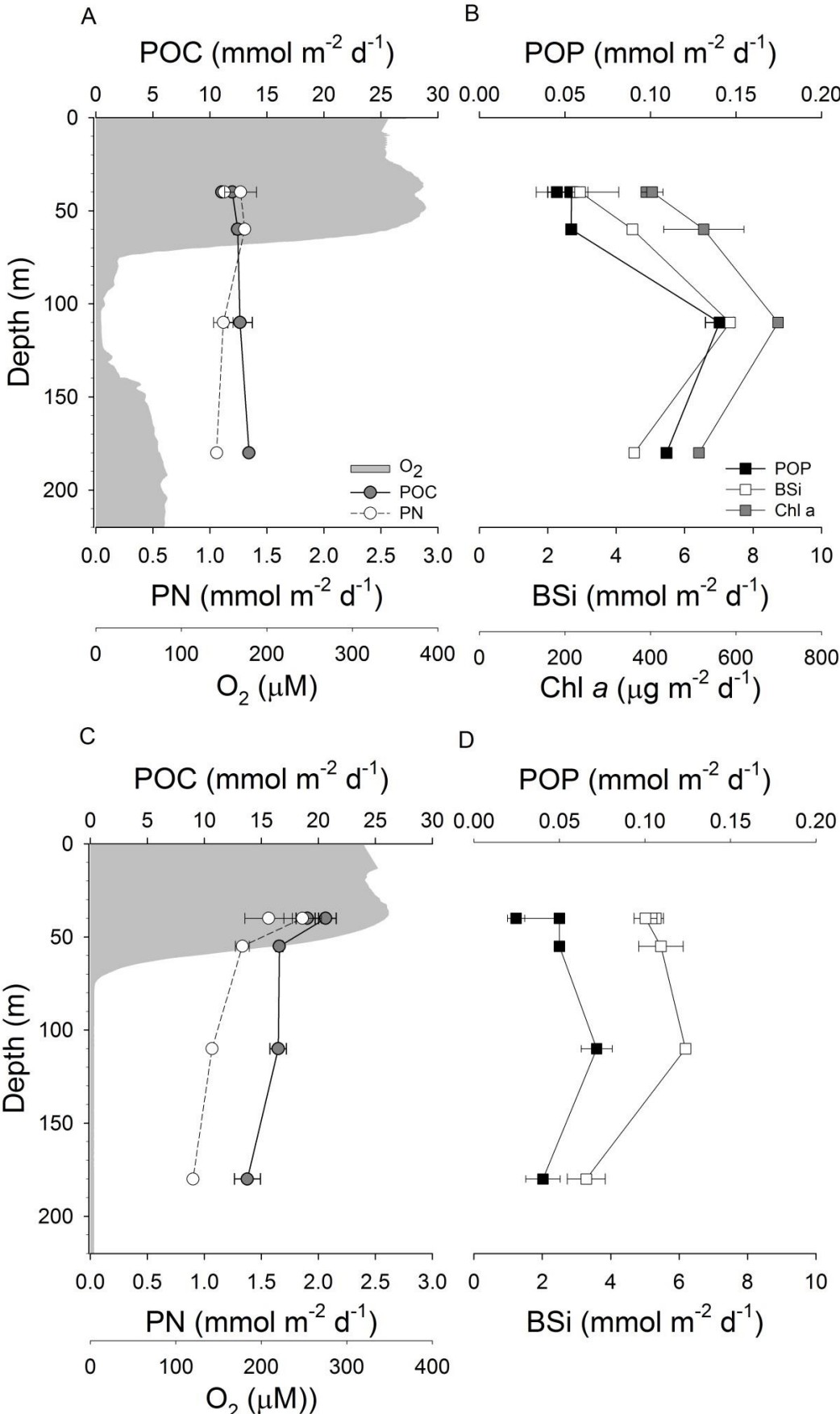

Fig. 5

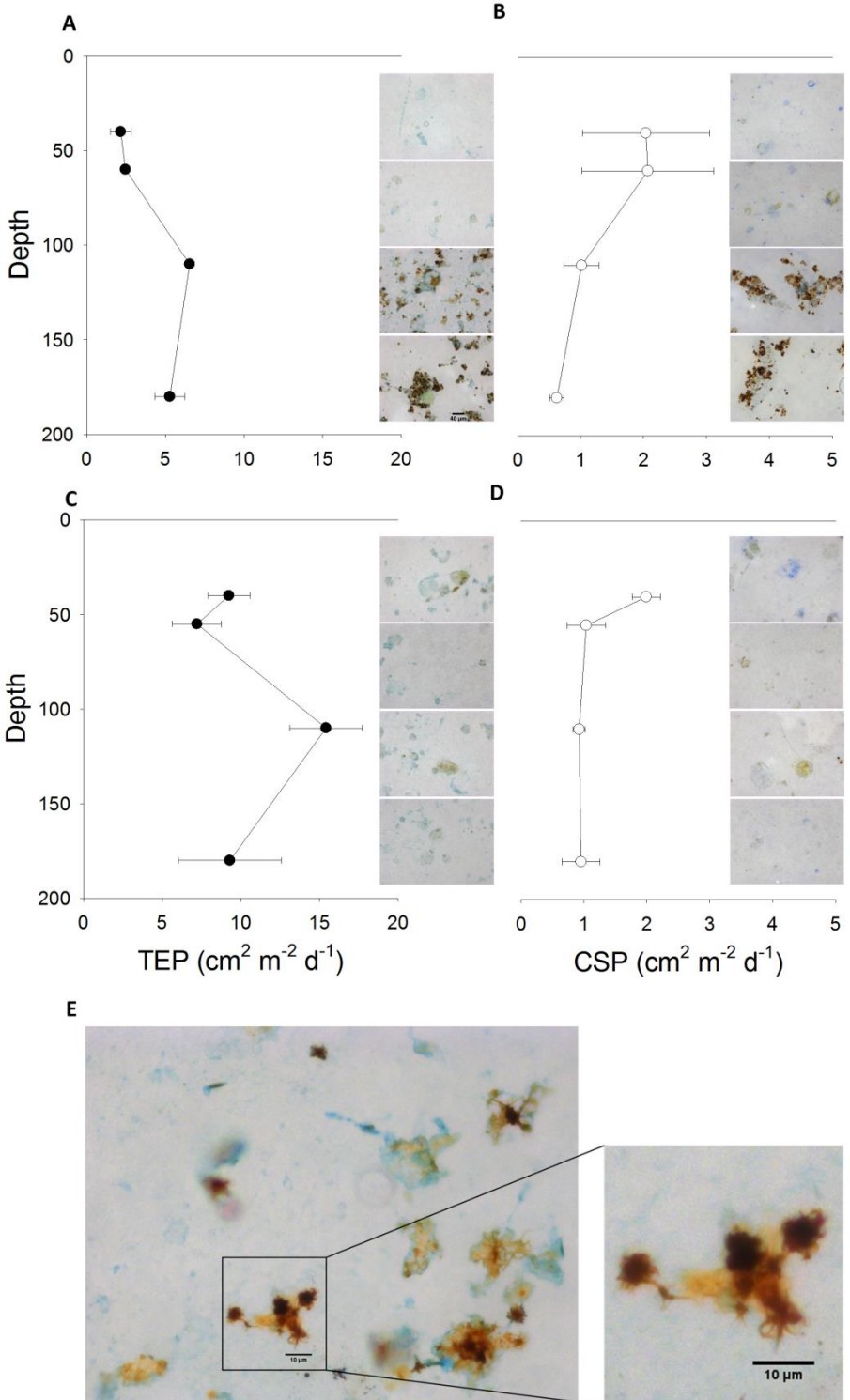

Fig. 6

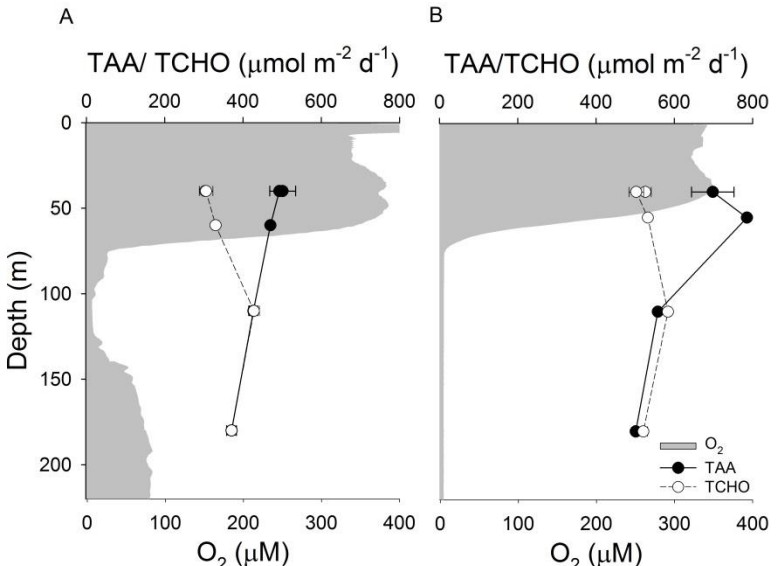

Fig. 7