# Peer review of "Composition and Vertical Flux of Particulate Organic Matter to the Oxygen Minimum Zone"

_Biogeosciences, 2018_

## Short Comment (SC1) · 3 Sep 2018

The authors demonstrate their investigations of particulate matter in the water column and in sediment traps in two basins (Gotland Basin (GB) and Landsort Deep (LD)) to estimate its composition and the particle flux in theses basins and how it changes dependent if the deep water is oxygenated or not. Thus, the manuscript can be a significant contribution to understand the biogeochemical processes and its spatial variation in the Baltic Sea. However, the ms is difficult to read and to get a 'take home massage' because it contains many nonspecific verbalization. For example, it is often unclear

which depth horizons are meant or why the situation at a certain depth is mentioned at this place (e.g. 110m in line 31). The influence of the MPI must be presented more clearly and substantiated. This requires comparison of a defined depth range which is oxygen rich in one basin and low in oxygen in the other. In the GB, the situation after MBI should be compared with a situation during a stagnation period, if possible. Many of these things are present in the ms but not clearly and focused demonstrated. I would expect from the title what is transported from the overlaying layers to the OMZ. Indeed, data for the whole water column of the GB and up to 200m in the LD are given without any Accentuations.

Abstract: The objectives are not clear Line 17-18: "Oxygen (O2) depletion may improve the efficiency of the biological carbon pump". Is this sentence the hypothesis of the work? If yes, than it has to be indicated and it needs to be answered at the end of the abstract.

Line 20-21: I would replace "Major Baltic Inflow "by" salt water inflow" here.

Line 24-32: It is difficult to understand what the comparison of the two depths means for the task.

Line 29: Why "contrastingly"? POC and PN decreased too. Line 33: why "may form"? I think it is a result of this work.

Line 38: sink instead of sank

Introduction: Line 78-80: " …..(Tamelander et al. 2017)". Please replace "On the long term, a decrease in OM downward flux may limit the oxygen depletion." By "The reduction of nutrient inputs as target by HELCOM can cause a decrease in OM downward flux and limit the oxygen depletion."

Line79-80. I would delete the last sentence of the chapter.

Line 88: "… Carstensen et al. 2014)". Recommendation: Salt water inflows from the North Sea…….

Line 91-92 "Saltier, denser, O2-rich North Sea waters entered the western Baltic Sea in December 2014 and reached the Gotland Basin on February 2015." The sentence could be deleted.

Line 92-95: "At the time of sampling, this MBI also affected the neighboring Faroe Deep; but not the LD, located further northwest." The sentence can be deleted.

Line 100: Please add the water depth in which the redox lines occur.

Line 99-100: This chapter can be shortened and combined with the chapter before.

Line 118-119: "enriched with OM; specifically with transparent exopolymer particles".

Line 134-142: A clear objective is missing here.

Methods Line 146: "surface-tethered sediment traps"that's not true. Traps were also installed in 180 m

Line 147-148: depth of water sampling should be given here.

Line 150: conductivity temperature depth? Suggestion: Temperature, salinity and O2 concentration were determined at each station using a Sea-Bird (CTD) probe equipped with a oxygen (Oxyguard, PreSens) sensor that was calibrated…

Line 155: According to Table 1, there are 3 or 4 depths in which the traps were exposed. That should be mentioned here as well.

Line 174-178: Ammonium has to be measured in an unfiltered sample. However if samples for nutrient analysis are stored frozen and analyzed using an auto-analyzer, then filtration is necessary. Please correct.

Line 191-192: Please add the wave lengths

Line 215-219: Suggestion: "Particle number and area was measured semi-automatically using an image analysis system including the WCIF ImageJ software. Image analysis of TEP and CSP were and conducted after Engel (2009). Additionally,

TEP and CSP in water samples from the stations where we deployed sediment traps were analysed spectrophotometrically according to Passow and Alldredge (1995) and Cisternas‐Novoa et al. (2014) respectively. Why was the additional method applied?

Line 223: For TEP and CSP it should be mentioned that the red and the green channel were used?. Here should only be mentioned that the blue channel was used.

Line 226 and line 233: Please delete the word "directly". When storing samples, there is no direct measurement.

Results Generally: I suggest that the results should be demonstrated for the basins successively (at first for the GB and after it for the LD) and not switched between the basins. In the vertical direction it should be started with the surface then successively the deeper layers whereby the depth of each layer should be defined to understand the results reported thereafter. Information about temperature and salinity is missing in the text.

Line 250-251: Information about the thermocline should be moved to the beginning of the chapter. The traps were exposed for one or two days. The water samples were taken at the same time. I don't believe that there was such a large rage where the thermocline was located during this short time.

Line 259: Which depth is meant with surface water?

Line 260: Suggestion for changes: (6 $\mu$M at 80 and 140 m, and 0.12 260 $\mu$M at 110 m). It could be added already here that the upper (80m) and lower (140m) bounds and 110m the core of the OMZ are. It is mentioned later, but it should be already included here.

Line 270: Because the conditions in water column are reported, it should be mentioned that nitrite had a maximum at 370m (Fig.2).

Line 273-274: To which depth the second nutrient values apply; "the upper 110m" is

confusing here.

Line 275: 0.22$\mu$M?

Line 269-276: The individual nutrients should be described one after the other and not switch between them several times.

Line 285: Please insert some data.

Line 286: Do you mean the sum of pico- and nanophytoplankton?

Line 287-288: 92% of what? Recommentation: Picocyanobacteria determined by phycoerythrinfluorescence amounted 92% in GB and 96% in LB of the total picophytoplankton and was 30% . . .. .

Line 289: "The abundance of larger phytoplankton (>5$\mu$m) was determined by microscopy". The sentence can be deleted. It is described in the methods.

Line 293: Filamentous unicellular cyanobacteria. A cyanobacteria filament always consists of more than one cell.

Line 292-293: "Cyanobacteria were 60% less abundant in the GB than in the LD." It is mentioned 2 sentences before. It can be deleted here.

Line 297: 95% of what

Line 302: which layer is meant with the surface.

Line 316: . . ..decreased quickly at 10m. . ." Rather: . . .decreased quickly below 10m. . ..

Line 318: The units of TEP and CSP should be explained in the methods.

Line 324: ". . .were only observed. . ." instead of ". . ..were observed. . ."

Line 331: What is ESD? Please give the full name.

Line 341: "POC flux slightly increased by 18% from the shallowest (40 m) to the deepest (180 m) depth. Fluxes of PN and CSP were higher at 40 and 60 m and decreased

by 19 and 70% from 60 to 180 m.. . ..".  I assume the layers 0-40m and 0-180m are meant.

Line 356: ". . ..sediment traps at 110 m and 180 m. MnOx- like were They occurred as single particles and forming formed . . ... and with other particles. . .. ".

Line 361: ". . ...ranged from 0.6 to 16.5 mm (media mean 1.8) at 110 m (Table 4).

Line 371-381: These chapters would be easier to understand if data are inserted.

Line 390: Please indicate in the method chapter how the DI has been calculated. Line 300-401: "We assess the potential influence of increased O2 concentration caused by the 2014/2015 MBI in the GB on the chemical composition and degradation stage of the sinking and suspended OM relative to the anoxic LD." In my view, this is not clear enough in the ms, including the discussion.

Discussion The discussion involves a lot of repetition of the results.

Line 404-405 "...primary production". Do you mean phytoplankton biomass? PP measurements were not included in this study.

Line 410-411: "Pico-phytoplankton cell abundance (cell mL-1) dominated the small phytoplankton size fraction $< 5\mu$m (Table 2), suggesting a significant contribution to PP and Chl a concentration. This can not be deduced from the abundance alone.

Line 421-422: "Cell abundance of total phytoplankton ($>5\ \mu$m) were not significantly different (p=0.74) in the GB and the LD." Which phytoplankton group refers to this statement. According to Table 3 the cell counts in both basins differed. I am wondering that the differences are not significant.

Line 434: "Our samples were collected right after the peak of the spring bloom. . ...". That is not right.  The spring bloom occurs, for example in the Gotland Basin, from the middle until the end of April (see also publications by B. Schneider et al.).  The investigations were carried out in June.

Line435-437: "...TEP concentrations had not reached the usually higher summer value yet since phosphate remained present in the water column (potentially not limiting the PP)". Please make the relationship more clear.

Line 444: It should be noted at what depth the OMZ was located before the salt water inflow. Recommendation for rewriting the sentence: The MBI changed the vertical distribution of O2 in the GB by increasing its concentrations in depth below...m and relocation of the oxygen deficient layers from ..m to 74-140 m depth.

Line 452-453: "MBIs can have a major impact on nutrient recycling". Such general statements should be reduced throughout the ms.

Line 480: ".... Carbon flux below the euphotic zone...". To the bottom or to what depth?

Line 485-486: "....the estimations based on our results from the GB are higher than the C fluxes predicted by those models." Here it should be taken into account that the measurements are obtained only from a single measurement over one or two days. The question is how representative a single measurement is.

The subsequent paragraphs and chapter should be focused. At the moment it is very diffuse and the message is not clear.

Table 3: It is not clear for me how the filamentous cyanobacteria were counted. Were the single cells in the filaments counted or were it counted as units of $50\mu$m or $100\mu$m length, as it is usually performed.

Fig.2A: The scale of the x-Axis for salinity is wrong.

In Fig. 4: It seems that only one or two depth are sampled. It should be indicated by zero-values if all depth are investigated and no particle is found.

Recommendation: Moderate revision

---

## Referee Comment (RC2) · T. Jilbert (Referee) · 15 Oct 2018

In this study the authors use a wide range of analyses to investigate the vertical structure of suspended and sinking particulate matter composition in two stratified basins of the Baltic Sea following the MBI of 2014-2015. The dataset is large and interesting, but I concur with the first reviewer's assessment that the study lacks a clear focal message. For this reason I would encourage the authors to streamline the text when making their revisions.

My principal scientific comment about the paper would be that the authors have not acknowledged the possibility that vertical profiles of dissolved and particulate constituents in the Gotland Basin may be influenced by displacement effects. Following the MBI of 2014-2015, the sub-halocline water column of the GB experienced significant turbulent mixing between 'old' and 'new' water masses. A lot of the changes in water chemistry that occurred during 2015 were caused by displacement of old, stagnant water by water masses associated with the MBI (see e.g. Myllykangas et al., ESD 8, 2017). For example, the low concentrations of Si(OH)4 and PO4 in the deepest samples of the GB (Fig. 2A) are very likely due to enhanced contribution of oxic, low-nutrient water at this depth, and not due to scavenging of these constituents onto MnOx particles as suggested by the authors for phosphate (Line 464 and in the Conclusions). Displacement may have also influenced the vertical structure of suspended and sinking particulate matter, so this angle should be included when interpreting the results.

In addition I would urge the authors to check their text thoroughly for typographic, spelling and grammatical errors. I have highlighted a few in my minor comments but there are likely several more.

Kind regards,

Tom Jilbert

Minor comments

Line 61: spelling: "allochthonous"

Line 95: spelling and grammar: the correct spelling is "Fårö"; Use "In the LD" rather than "At the LD"

Line 110: rephrase (difficult to understand)

Line 156: grammar: Use "consisted of" rather than "consisted in"

Line 166: what is the meaning of "caped"?

Line 181: grammar: Use "in duplicate" rather than "in duplicated"

Line 220: rephrase (difficult to understand)

Line 321: spelling "below"

Line 354: what is the meaning of "and similar to the water column"?

Line 356: word missing: "MnOx like were…"

Line 357: Remove colon (:) before "TEP"

Line 358: Define ESD

Line 362: Avoid staring a sentence with an acronym

Line 375: add space before bracket. Also "Redfield's" should be "Redfield ratio"

Line 390: DI should be introduced and defined in the Methods section

Line 432: grammar: "may be enhanced"

Line 437: typographic errors

Line 444: typographic errors

Line 451: "compounds" plural

Line 453: spelling: "phosphorus"

Line 464: Rephrase and check grammar, tenses, etc.

Line 468-470: these statements belong in Results rather than Discussion

Line 489-90: typographic errors

Line 519: Mn2+ is not an electron acceptor

Line 526: PN and CSP are not compounds. Rephrase.

Line 597: Nisken bottle, not CTD

Table 2: should the units be "cells/mL)"?

Fig. 4: are these all the sampling depths for MnOX-like particles? If samples from other depths were studied but yielded zero particles, these should also be included in the plot

---

## Author Comment (AC1) · 9 Nov 2018

We thank the referee for her helpful observations, comments, and suggestions. Detail answers to the comments are provided below and in the attached pdf.

The authors demonstrate their investigations of particulate matter in the water column and in sediment traps in two basins (Gotland Basin (GB) and Landsort Deep (LD)) to estimate its composition and the particle flux in theses basins and how it changes dependent if the deep water is oxygenated or not. Thus, the manuscript can be a

significant contribution to understand the biogeochemical processes and its spatial variation in the Baltic Sea. However, the ms is difficult to read and to get a 'take home massage' because it contains many nonspecific verbalization. For example, it is often unclear which depth horizons are meant or why the situation at a certain depth is mentioned at this place (e.g. 110m in line 31). The influence of the MPI must be presented more clearly and substantiated. This requires comparison of a defined depth range which is oxygen rich in one basin and low in oxygen in the other. In the GB, the situation after MBI should be compared with a situation during a stagnation period, if possible. Many of these things are present in the ms but not clearly and focused demonstrated. I would expect from the title what is transported from the overlaying layers to the OMZ. Indeed, data for the whole water column of the GB and up to 200m in the LD are given without any Accentuations. Abstract: The objectives are not clear Line 17-18: "Oxygen (O2) depletion may improve the efficiency of the biological carbon pump". Is this sentence the hypothesis of the work? If yes, than it has to be indicated and it needs to be answered at the end of the abstract. AR: We will add information to the abstract, about the oxygen conditions at each depth where we deployed a sediment trap in GB and LD (ex: oxygenated surface, the core of the oxygen minimum zone and deep water oxygenated by the inflow) to clarify why we mention specific depths. We explicitly add the hypothesis and rephrase the objectives for clarity. We answered the hypothesis at the end of the abstract when we expose the differences in POM and export between the two studied stations, and we propose a mechanism to explain those differences. Information about oxygen conditions at depths where sediment traps were deployed in each station was added in L22-25. We explicitly add the hypothesis to the abstract (L25-27) Line 20-21: I would replace "Major Baltic Inflow "by" salt water inflow" here. AR: This was change to "a major oxygen-rich saltwater inflow" (L21) Line 24-32: It is difficult to understand what the comparison of the two depths means for the task. AR: This paragraph was modified for clarity and as mentioned above, the oxygen conditions at each depth are mentioned (ex: oxygenated surface, the core of the oxygen minimum zone and deep water oxygenated by the

inflow) Line 29: Why "contrastingly"? POC and PN decreased too. AR: We agree this was confusing "contrastingly" was deleted Line 33: why "may form"? I think it is a result of this work. AR: We said "may form" because, as we explain in the next line, our results suggest that MnOx-like particles aggregate with POM, this is what we propose. However, we think that to ensure that MnOx and POM aggregate and the specific composition of those aggregates, we will need to investigate aggregate formation and composition further. Line 38: sink instead of sank AR: This has been fixed. Introduction: Line 78-80: " . . ..(Tamelander et al. 2017)". Please replace "On the long term, a decrease in OM downward flux may limit the oxygen depletion." By "The reduction of nutrient inputs as target by HELCOM can cause a decrease in OM downward flux and limit the oxygen depletion." AR: The sentence has been replaced according to the reviewer suggestion. Line79-80. I would delete the last sentence of the chapter. AR: The last sentence of the paragraph "However, to fully suppress hypoxia enhanced ventilation would be necessary the bottom waters of the Baltic Sea" is to emphasize that the bottom-water oxygen concentrations, are not only controlled by the nutrient loads, but also by physical factors like the frequency and intensity of the saltier water inflow, which had a decadal variability and it is modulated by meteorological forcing (e.g., Carstensen et al. 2014). This is explained in the next paragraph. Line 88: ". . . Carstensen et al. 2014)". Recommendation: Salt water inflows from the North Sea. . .. . . AR: The sentence has been modified according to the reviewer recommendation. Line 91-92 "Saltier, denser, O2-rich North Sea waters entered the western Baltic Sea in December 2014 and reached the Gotland Basin on February 2015." The sentence could be deleted. Line 92-95: "At the time of sampling, this MBI also affected the neighboring Faroe Deep; but not the LD, located further northwest. The sentence can be deleted. AR: We modified this paragraph and combined those sentences as "At the time of sampling, this MBI had reached the Gotland Basin, but did not affect the LD, located further northwest." to emphasize that the MBI oxygenated the deep waters of GB, but not those of LD (L97-98). Line 100: Please add the water depth in which the redox lines occur. AR: We added the redoxcline depth from literature (between 120

and 150 m depth; L102). In the results, we define the redoxcline depth determined from the O2 and H2S concentrations during our sampling. Line 99-100: This chapter can be shortened and combined with the chapter before. AR: We think that provide background information on the chemical reactions occurring in the redoxcline, and how previous MBI had altered redox conditions, is important to frame our idea that changes in oxygenation enhance the formation of MnOx that aggregate with POM and alter POM distribution and export Line 118-119: "enriched with OM; specifically with transparent exopolymer particles". AR: This has been fixed (L122). Line 134-142: A clear objective is missing here. AR: This paragraph has been modified to clarify the objectives of the study (L138-146) Methods Line 146: "surface-tethered sediment traps"that's not true. Traps were also installed in 180 m AR: As explained in the section "Sediment trap design and deployment"(L158-164) and in more detail in Engel et al. 2017 and Knauer et al. 1979, our traps consisted of 12 particle interceptor tubes (PITs) framed in five PVC crosses. Crosses with PITs tubes are attached at four depths: 2 at 40m, 60m, 110m, and 180m. The entire array is attached to the flotation gear that consists on a polypropylene line attached to two different types of flotation spheres. The entire flotation array is secured to the surface spar (a large yellow buoy) on which the flashlight and positioning systems are mounted. Line 147-148: depth of water sampling should be given here. AR: Water sampling depths are in table 2; this was added to the text (L152) Line 150: conductivity temperature depth? Suggestion: Temperature, salinity and O2 concentration were determined at each station using a Sea-Bird (CTD) probe equipped with a oxygen (Oxyguard, PreSens) sensor that was calibrated. . . AR: This paragraph has been modified according to the reviewer recommendations (L153-154) Line 155: According to Table 1, there are 3 or 4 depths in which the traps were exposed. That should be mentioned here as well. AR: The sediment trap depths were added to the text (L158). Line 174-178: Ammonium has to be measured in an unfiltered sample. However if samples for nutrient analysis are stored frozen and analyzed using an auto-analyzer, then filtration is necessary. Please correct. AR: We thank the reviewer for this observation. The paragraph was corrected

(L176-181). Line 191-192: Please add the wave lengths AR: The wavelengths were added to the text (440/685 nm; L195) Line 215-219: Suggestion: "Particle number and area was measured semiautomatically using an image analysis system including the WCIF ImageJ software. Image analysis of TEP and CSP were and conducted after Engel (2009). Additionally, TEP and CSP in water samples from the stations where we deployed sediment traps were analysed spectrophotometrically according to Passow and Alldredge (1995) and Cisternas' RNovoa et al. (2014) respectively. Why was the additional method ap- ĚĞ plied? AR: We modified the sentence according to the reviewer suggestion (L219-222). The additional spectrophotometrically method was used to measured TEP and CSP concentration in the water column. Since the spectrophotometric method is less labor intensive, it allows for sampling of the water column with higher vertical resolution (between 9 and 12 depths; L223) Line 223: For TEP and CSP it should be mentioned that the red and the green channel were used?. Here should only be mentioned that the blue channel was used. AR: We modified the TEP, CSP (L219-222) and MnOx (L230-232) image analysis section according to the reviewer suggestion. Line 226 and line 233: Please delete the word "directly". When storing samples, there is no direct measurement. AR: We deleted the word "directly" Results Generally: I suggest that the results should be demonstrated for the basins successively (at first for the GB and after it for the LD) and not switched between the basins. In the vertical direction it should be started with the surface then successively the deeper layers whereby the depth of each layer should be defined to understand the results reported thereafter. Information about temperature and salinity is missing in the text. Line 250-251: Information about the thermocline should be moved to the beginning of the chapter. The traps were exposed for one or two days. The water samples were taken at the same time. I don't believe that there was such a large rage where the thermocline was located during this short time. AR: The organization of this paragraph was changed according to the reviewer suggestion, i.e. Temperature, salinity and O2 conditions were discussed first in GB and then in LD. Information about temperature and salinity were added to the text. We include the thermocline

information based on the measurements made during the deployment of the sediment traps. The depth range presented correspond to the initial and end depth were the temperature had a rapid decreased, this information was added to the text. Line 259: Which depth is meant with surface water? AR: We added to the text that surface water mean upper 10 m. Line 260: Suggestion for changes: (6 $\mu$M at 80 and 140 m, and 0.12 260 $\mu$M at 110 m). It could be added already here that the upper (80m) and lower (140m) bounds and 110m the core of the OMZ are. It is mentioned later, but it should be already included here. AR: We added the information about upper, lower oxycline and OMZ core in this sentence (L278-279). Line 270: Because the conditions in water column are reported, it should be mentioned that nitrite had a maximum at 370m (Fig.2). AR: We added the maximum of nitrite at 250 m and nitrate at 400 m to the text (L291-292) Line 273-274: To which depth the second nutrient values apply; "the upper 110m" is confusing here. Line 275: 0.22$\mu$M? Line 269-276: The individual nutrients should be described one after the other and not switch between them several times. AR: We modified the nutrient section for clarity Line 285: Please insert some data. Line 286: Do you mean the sum of pico- and nanophytoplankton? Line 287-288: 92% of what? Recommentation: Picocyanobacteria determined by phycoerythrinfluorescence amounted 92% in GB and 96% in LB of the total picophytoplankton and was 30% . . . ... AR: We modified this paragraph according to the reviewer suggestion (L310-311) Line 289: "The abundance of larger phytoplankton (>5$\mu$m) was determined by microscopy". The sentence can be deleted. It is described in the methods. AR: This sentence was modified (L313) Line 293: Filamentous unicellular cyanobacteria. A cyanobacteria filament always consists of more than one cell. A: We thank the reviewer for this observation, "unicellular" was deleted Line 292-293: "Cyanobacteria were 60% less abundant in the GB than in the LD." It is mentioned 2 sentences before. It can be deleted here. AR: We deleted this sentence Line 297: 95% of what AR: Sorry, we do not understand this comment Line 302: which layer is meant with the surface. AR: With surface we referred to the upper 10 m of the water column. This was added to the text. Line 316: . . ..decreased quickly at 10m. . ." Rather: . . .decreased quickly

below 10m. . .. AR: We thank the reviewer for this observation, the sentence was fixed
Line 318: The units of TEP and CSP should be explained in the methods. AR: We add
the units for TEP and CSP to the method section (L225-228) Line 324: ". . .were only
observed. . ." instead of ". . ..were observed. . ." AR: This has been modified. Line
331: What is ESD? Please give the full name. AR: We added equivalent spherical
diameter to the text Line 341: "POC flux slightly increased by 18% from the shallowest
(40 m) to the deepest (180 m) depth. Fluxes of PN and CSP were higher at 40 and
60 m and decreased by 19 and 70% from 60 to 180 m.. . .". I assume the layers
0-40m and 0-180m are meant. AR: Yes, since we collected discrete samples at 40,
60, 110 and 180m, the assumption is that at each depth the sediment trap collected
the particles formed and sinking from the euphotic zone. Line 356: ". . ..sediment
traps at 110 m and 180 m. MnOx- like were They occurred as single particles and
forming formed . . ... and with other particles. . .. ". AR: This has been modified. Line
361: ". . ...ranged from 0.6 to 16.5 mm (media mean 1.8) at 110 m (Table 4). AR:
We referred to the "median" as a measure of central tendency. The word "median"
was fixed in the text Line 371-381: These chapters would be easier to understand if
data are inserted. AR: We added the POC: PN ratio range to make the paragraph
easier to follow. Line 390: Please indicate in the method chapter how the DI has been
calculated. AR: We added the DI calculation to the method section (L240-244) Line
300-401: "We assess the potential influence of increased O2 concentration caused
by the 2014/2015 MBI in the GB on the chemical composition and degradation stage
of the sinking and suspended OM relative to the anoxic LD." In my view, this is not
clear enough in the ms, including the discussion. Discussion The discussion involves
a lot of repetition of the results. Line 404-405 "...primary production". Do you mean
phytoplankton biomass? PP measurements were not included in this study. AR:
We replaced PP by phytoplankton biomass Line 410-411: "Pico-phytoplankton cell
abundance (cell mL-1) dominated the small phytoplankton size fraction $< 5\mu m$ (Table
2), suggesting a significant contribution to PP and Chl a concentration. This can not
be deduced from the abundance alone. AR: We deleted the sentence "suggesting a

significant contribution to PP and Chl a concentration" Line 421-422: "Cell abundance of total phytoplankton (>5 $\mu$m) were not significantly different (p=0.74) in the GB and the LD." Which phytoplankton group refers to this statement. According to Table 3 the cell counts in both basins differed. I am wondering that the differences are not significant. AR: To determine if the total phytoplankton abundances (considering all groups presented in table 3) were significantly different we used the Mann-Whitney U-test. The p-value (0.74) indicates that there is not sufficient evidence to indicate that the medians of those two data set were significantly different. Line 434: "Our samples were collected right after the peak of the spring bloom. . ...". That is not right. The spring bloom occurs, for example in the Gotland Basin, from the middle until the end of April (see also publications by B. Schneider et al.). The investigations were carried out in June. AR: We corrected this paragraph. Line435-437: ". . .TEP concentrations had not reached the usually higher summer value yet since phosphate remained present in the water column (potentially not limiting the PP)". Please make the relationship more clear. AR: We modified this paragraph to make clear that even though we sample in June, the high Chla concentration and the phosphate still present in the water column could indicate that PP was not nutrient limited yet. The presence of nutrients (phosphate) may be an explanation of why TEP concentration was lower than reported before for summer in the Baltic Sea when PP was low due to nutrient limitation. Line 444: It should be noted at what depth the OMZ was located before the salt water inflow. Recommendation for rewriting the sentence: The MBI changed the vertical distribution of O2 in the GB by increasing its concentrations in depth below...m and relocation of the oxygen deficient layers from ..m to 74-140 m depth. AR: We modified the text according to the reviewer recommendations Line 452-453: "MBIs can have a major impact on nutrient recycling". Such general statements should be reduced throughout the ms. AR: This sentence was moved to the beginning of the paragraph since was introducing the effect of MBI in nutrient distribution. Line 480: ". . .. Carbon flux below the euphotic zone. . .". To the bottom or to what depth? AR: For clarification, this sentence was modified to "Our measurement of carbon flux at

40 m , below the euphotic zone, were..." Line 485-486: ". . ..the estimations based on our results from the GB are higher than the C fluxes predicted by those models." Here it should be taken into account that the measurements are obtained only from a single measurement over one or two days. The question is how representative a single measurement is. The subsequent paragraphs and chapter should be focused. At the moment it is very diffuse and the message is not clear. AR: We agreed with the reviewer that our study represent only one discrete measurement; however, the objective of mention the results of previous estimations from modeling studies was precisely to add some context to our results. We re-organize the discussion to make it clearer and more focus. Table 3: It is not clear for me how the filamentous cyanobacteria were counted. Were the single cells in the filaments counted or were it counted as units of $50\mu$m or $100\mu$m length, as it is usually performed. AR: The filamentous cyanobacteria were counted as single filament as it is usually performed. The word unicellular was deleted from the table. Fig.2A: The scale of the x-Axis for salinity is wrong. AR: We thank the reviewer for this observation; the salinity scale in figure 2a is fixed In Fig. 4: It seems that only one or two depth are sampled. It should be indicated by zero-values if all depth are investigated and no particle is found. AR: We added all values to figure 4, included the depths with zero particles.

Please also note the supplement to this comment:
https://www.biogeosciences-discuss.net/bg-2018-360/bg-2018-360-AC1-supplement.pdf

―――――――――――――――――――――――

---

## Author Comment (AC2) · 9 Nov 2018

We thank Dr. Tom Jilbert for reviewing our manuscript. Detail answers to the comments are provided below and in the attached pdf.

In this study the authors use a wide range of analyses to investigate the vertical structure of suspended and sinking particulate matter composition in two stratified basins of the Baltic Sea following the MBI of 2014-2015. The data set is large and interesting, but I concur with the first reviewer's assessment that the study lacks a clear

focal message. For this reason I would encourage the authors to streamline the text when making their revisions. My principal scientific comment about the paper would be that the authors have not acknowledged the possibility that vertical profiles of dissolved and particulate constituents in the Gotland Basin may be influenced by displacement effects. Following the MBI of 2014-2015, the sub-halocline water column of the GB experienced significant turbulent mixing between 'old' and 'new' water masses. A lot of the changes in water chemistry that occurred during 2015 were caused by displacement of old, stagnant water by water masses associated with the MBI (see e.g. Myllykangas et al., ESD 8, 2017). For example, the low concentrations of $Si(OH)_4$ and $PO_4$ in the deepest samples of the GB (Fig. 2A) are very likely due to enhanced contribution of oxic, low-nutrient water at this depth, and not due to scavenging of these constituents onto $MnO_x$ particles as suggested by the authors for phosphate (Line 464 and in the Conclusions). Displacement may have also influenced the vertical structure of suspended and sinking particulate matter, so this angle should be included when interpreting the results. In addition I would urge the authors to check their text thoroughly for typographic, spelling and grammatical errors. I have highlighted a few in my minor comments but there are likely several more. Kind regards, Tom Jilbert AR: We agreed with the reviewer that the displacement effects associated with the 2104/2015 MBI might have influenced the vertical profiles of dissolved and particulate constituent. Therefore, we add this aspect to the discussion (L478-484) of the vertical profile of nutrients (L504-510), vertical profile of particulate organic in the water column, and particulate organic matter fluxes (L554-558). However, even when we acknowledge that the net effect of the MBI in the particulate organic matter (POM) distribution and export efficiency is a combination of physical effects and biogeochemical changes; this does not modify our conclusion. Our results suggest that changes in the water chemistry related to the MBI and the consequent transport or in-situ formation of $MnO_x$ due to the favorable redox conditions may impact the distribution, degradation, and of export of POM in the GB We thank Dr. Tom Jilbert for his useful comments and corrections. We fixed all the mistakes pointed out and

carefully revised the manuscript to avoid future spelling and grammatical errors. Minor comments Line 61: spelling: "allochthonous" AR: We corrected the spelling mistake. Line 95: spelling and grammar: the correct spelling is "Fårö"; Use "In the LD" rather than "At the LD" AR: We corrected the grammar mistake Line 110: rephrase (difficult to understand) AR: We rephrased the sentence. Line 156: grammar: Use "consisted of" rather than "consisted in" AR: We changed the preposition Line 166: what is the meaning of "caped"? AR: We changed the word "capped" for "covered" Line 181: grammar: Use "in duplicate" rather than "in duplicated" AR: It has been fixed Line 220: rephrase (difficult to understand) AR: We rephrased the sentence Line 321: spelling "below" AR: We fixed the spelling of "below" in the ms Line 354: what is the meaning of "and similar to the water column"? AR: For clarity, we modified this sentence to "similar to the water samples," Line 356: word missing: "MnOx like were..." AR: We fixed the sentence Line 357: Remove colon (:) before "TEP" AR: We removed the colon Line 358: Define ESD AR: We added the definition of equivalent spherical diameter (ESD) to the text. Line 362: Avoid staring a sentence with an acronym AR: This has been fixed Line 375: add space before bracket. Also "Redfield's" should be "Redfield ratio" AR: We fixed those mistakes Line 390: DI should be introduced and defined in the Methods section AR: We added the definition and calculation of the DI to the method section. Line 432: grammar: "may be enhanced" AR: We changed this line Line 437: typographic errors AR: We fixed the typographic error Line 444: typographic errors AR: We fixed the typographic error Line 451: "compounds" plural AR: We corrected the word to "compounds" Line 453: spelling: "phosphorus" AR: We corrected the spelling of "phosphorus" in the ms. Line 464: Rephrase and check grammar, tenses, etc. AR: We rephrase and corrected the grammar of this paragraph. Line 468-470: these statements belong in Results rather than Discussion AR: We moved the statement to the results section Line 489-90: typographic errors AR: We modified this paragraph and fixed the errors. Line 519: Mn2+ is not an electron acceptor AR: We fixed this mistake Line 526: PN and CSP are not compounds. Rephrase. AR: We changed "compounds" to "components of POM" Line 597: Nisken bottle, not CTD AR: We

replaced CTD by Niskin bottle Table 2: should the units be "cells/mL)"? AR: We modified how we showed the units to (cell ml-1) Fig. 4: are these all the sampling depths for MnOX-like particles? If samples from other depths were studied but yielded zero particles, these should also be included in the plot AR: We added all values to figure 4, included the depths with low abundance or zero particles

Please also note the supplement to this comment:
https://www.biogeosciences-discuss.net/bg-2018-360/bg-2018-360-AC2-supplement.pdf

---

## Editor Decision (ED1)

BG-2018-360 decision

Dear Carolina Cisternas-Novoa,

First of all I would like to thank both reviewers again for their insightful reviews and you for your better organized comments. I think the manuscript has improved significantly, but do still have some smaller and perhaps bigger issues. I guess one of the more major comments has to do with your MnOx-like particles, they are MnOx-like up until page 31 and then they become MnOx? Looking at your M&M section, I am not entirely clear what they are. If the method you describe for measurement of these particles is known to target MnOx particles I would expect one or more references indicating this. If not, than how do you know what these particles are? If they do contain MnOx, please indicate how you know, is MnOx-like the best name for them? MnOx containing might be better?

What is CSP? I know what it stands for, but what is it? Proteins or particle containing relatively large amounts of proteins? Is CSP correlated to DI?

Minor comments:
Line 16: "Sinking particles are the main form in which photosynthetically fixed carbon is transported from the euphotic zone to the ocean interior, the so called biological pump (BCP)". And of course you can discuss the Baltic being an ocean.
Line 22: GB, but not
Line 25: oxygenated by the inflow of relatively saline waters from the North Sea?
Line 28: POC has not been defined. Abbreviations and acronyms are typically defined the first time you use them in the abstract and again in the main text. If you only use them once, just use the full name.
Line 48: We're in the main text now so define BCP again.
Line 49: POC is defined, great, do the same in the abstract.
Line 53-54: What do these authors mean with "higher refractory nature of sinking particles in the OMZ?
Line 58-60: Didn't you just give that information?
Line 70: Gotland Basin (GB)
Line 73: Define OM or even POM
Line 84: "… would be necessary the bottom water…."?
Line 85: Gotland Basin, used before in line 70, define GB in line 70.
Line 87: just Kattegat should do I think.
Line 98: GB, it has been defined now.
Line 105: "Water column stratification…"
Line 108: Does that redoxcline still exists during/after the MB?
Line 109: this pelagic redoxcline is the original or the second redoxcline from the previous sentence?
Line 114-115: "…under oxic conditions OR  in the presence of nitrate they react with O2 and …" They will get oxidized, but in the latter case there is no O2?
Line 127: TEP is or TEP particles are
Line 133: by Stokes law…
Line 161: special variability?
Line 173: filled up to 10 L? How much water did you have to add to get to 10 L. And why would you dilute your samples?
Line 174: swimmers were removed with a 500 μm mesh screen?
Line 185: Aliquots from the 10 L? So not only trapped material, but diluted trapped material?
Line 192: Did you define POP?
Line 200-209: Am I correct in thinking that everything between 5 and 20 μm is both counted by flow cytometer and microscopy? I am assuming the flow cytometer is counting cells containing chlorophyll and/or phycoerythrin?

Line 214: Explain what coomassie normally stains. Later on explain why that might be interesting.

Line 231-236: If this is a known method to measure particles containing MnOx, refer to these papers or methods. MnOx-like particles could basically be anything. Line 243-244 "…. Undergoes degradation…"

Line 272: The deepest point sampled in the LD (430 m)

Line 299: increased to 38.9

Line 301: the lowest concentration was not at 180 m?

Line 304: GB µg per liter, LD µg per liter and µM?

Line 304-312: There is overlap in your defined pico- and nanophytoplankton (< 20 µm) and large phytoplankton (> 5 µm)?

Line 314: Filament counts, so the actual biomass or even individual cell counts could be even way larger, right? This also means that the other spp are a percentage of the total **counts** or phytoplankton **counts**. Not of the total biomass or even cell counts. Completely dwarfed by the cyano's therefore, right?

Line 315: total counts, so you could wonder if they are indeed significant based on biomass or cell counts taking into account you counted filaments rather than cyano cells.

Line 317: total phytoplankton counts

Line 318: phytoplankton counts

Line 331: TEP particles were counted right? So shouldn't this be counts as well? Line 340: so here is a little indication of what CSP might be… gel like, the question what it is still remains.

Line 345: MnOx containing particles? But did you actually measure MnOx? Or just counted particles that looked and behave as MnOx?

Line 362-363: by 18%

Line 376: Fully oxygenated water depths

Line 384-387: IS the TAA related to CSP?

Line 398: SOM or DOM?

Line 406: indicate the Redfield ratio for Si.

Line 411: TCHO?

Line 427: biomass? Or counts?

Line 437: the overlap in size classes again?

Line 483: MnOx-like and in line 485 you are sure it is actually MnOx, what has changed?

Line 531-532: So in Glockzin et al they actually measure manganese?

Line 533: ($H_2S$), pretty sure there is $H_2O$.

Line 537-538: "… redox conditions favorable for the formation of MnOx resulting in the high MnOx flux measured …"

Line 547: containing or should it be like?

Line 590: we consider

Line 592: mixed

Line 598: mixed

Line 606: than rather than that?

Line 615: "… how similar the biogeochemical conditions were …"

Line 625: Hence the $N_2$ fixing cyano's?

Line 660: transporting solid material from the and ?

Line 662: for the comparison of POM

---

## Author Response (AR2)

*Dear Marcel van der Meer,*

*Thanks for providing us the opportunity to revise our manuscript further. We would also like to thank the two reviewers and you for your valuable comments and recommendations to improve our manuscript. We carefully revised and modified the document following your latest suggestions and comments. In addition, we made extra modifications to the text to improve the clarity and focus of the whole manuscript.*
*Please find the detailed response (AR, blue italic font) to your comments and the changes made to the manuscript (ACMS, blue bold font). The precise line in the previous and in the revised version of the manuscript where each change was made is provided as well. Additionally, we provide a new version with tracked changes (this file), and a pdf version of the new manuscript (separate file).*
*We believe that the revised version satisfactorily addresses your questions and concerns and hope that the manuscript is now acceptable to BG. Should you have any additional requests or questions, please do not hesitate to contact me.*

*We are looking forward to hearing from you.*

*Sincerely,*

*Carolina Cisternas-Novoa*

BG-2018-360 decision

Editor comment (EC): Dear Carolina Cisternas-Novoa,
First of all I would like to thank both reviewers again for their insightful reviews and you for your better organized comments. I think the manuscript has improved significantly, but do still have some smaller and perhaps bigger issues. I guess one of the more major comments has to do with your MnOx-like particles, they are MnOx-like up until page 31 and then they become MnOx? Looking at your M&M section, I am not entirely clear what they are. If the method you describe for measurement of these particles is known to target MnOx particles I would expect one or more references indicating this. If not, than how do you know what these particles are? If they do contain MnOx, please indicate how you know, is MnOx-like the best name for them? MnOx containing might be better?
What is CSP? I know what it stands for, but what is it? Proteins or particle containing relatively large amounts of proteins? Is CSP correlated to DI?

*Author Reply (AR):*

1. *Regarding terminology discrepancy MnOx-like particles vs. MnOx containing particles.*
   *MnOx containing particles have been previously identified in the chemocline of Gotland Basin (GB, Dellwig et al., 2010; Dellwig et al., 2018; Glockzin et al., 2014; Neretin et al., 2003) and Landsort Deep (LD, Glockzin et al., 2014; Dellwig et al., 2010). The maximum abundance of those particles, coincide with the maximum concentration of particulate Mn and is located at the depth were $O_2$ concentration is ≤ 20 $\mu M$ and above measurable $H_2S$ (Neretin et al., 2003).*
   *Commonly MnOx containing particles are identified based on their morphologies, size and their elemental composition which is confirmed using a scanning electron microscopy (SEM) and energy dispersive x-ray microanalysis (EDX) (Neretin et al., 2003; Glockzin et al., 2014; Dellwig et al., 2010, 2018)*

*In our study, we did not measure the elemental composition of the particles. Thus, we identified them as "MnOx-like particles" based on similar morphology, size, and association with organic matter (OM) as MnOx containing particles previously reported for the Baltic Sea (e.g., Neretin et al., 2003 and Glockzin et al., 2014). The vertical distribution of our MnOx-like particles in the water column and sediment traps showed maximum concentration at the GB when the $O_2$ concentration was 40 $\mu M$ or lower, and no particles were visualized in the filters after the $H_2S$ was measurable in the LD.*

*Moreover, the vertical profile of MnOx-like particles in the GB in our study coincides with the dynamics of MnOx containing particles following the 2014-2015 MBI described by Dellwing et al. (2018). Their results showed a remarkable deposition of MnOx containing particles in the water column profiles and in sediment traps material collected at 186 m. In addition, the maximum concentration of particulate Mn was measured in the water column at 128 and 233 m in July 2015, which agreed with the vertical profile of Mn-like particles in the water column and in the sediment trap material that we found in this study.*

*The evidence mentioned above, strongly suggests that the particles observed and quantified in our study correspond to the previously reported MnOx containing particles. However, since we did not perform a chemical analysis to ensure their exact elemental composition, and instead define those particles based in their morphology, size and OM association we call them "MnOx-like particles." We quantified and sized them using particle recognition on filters and imaging processing as described in the methods section and similar to the method used by Neretin et al. (2003).*

*We carefully checked the text to ensure the use the term "MnOx-like particles" consistently when we refer to the particles identify and described in this study and differentiate them from previously reported MnOx containing particles.*

- **Author's changes in manuscript (ACMS):**
  **We added a paragraph to the method section explaining how we identified and quantified MnOx-like particles L255-269:**
  **"MnOx-containing particles have been commonly identified based on their morphology, size and elemental composition, confirmed by scanning electron microscopy (SEM) and energy dispersive x-ray microanalysis (EDX) (Neretin et al., 2003; Glockzin et al., 2014; Dellwig et al., 2010, 2018). In this study, we did not measure the elemental composition of the particles. Thus, we identified them as "MnOx-like particles" based on similar morphology, size, and association with organic matter (OM) as MnOx-containing particles previously described in the Baltic Sea (eg., Neretin et al., 2003 and Glockzin et al., 2014). The abundance and size of MnOx-like particles were determined using particle recognition on filters and imaging processing similar to the method used by Neretin et al. (2003) but without the chemical composition analysis of the particles. For the image analysis, we used the same images as for TEP and CSP analysis and modified image analysis procedure described above as follows: thirty images per filter (200x) were analyzed semi-automatically using ImageJ software. After RGB split, the blue channel pictures were used to quantify MnOx-like particles in the water column and sediment traps. In this manner, the MnOx-like particles were clearly visible with a negligible disruption from TEP or CSP stained blue."**

**We added a paragraph explaining why the particles in our study, denominated "MnOx-like particles", may correspond to the previously reported MnOx containing particles L611-617:**
**"The high flux of POC at GB coincided with the appearance of dark, star-shaped particles that we defined as MnOx-like particles, particularly evident at GB (Fig. 6a,b, and e), but also present in LD.  Based on their morphology, size, and aggregation with OM, we propose that those particles correspond to MnOx-containing particles enriched in OM that have been previously described at GB (Neretin et al., 2003; Pohl et al., 2004; Glockzin et al., 2014; Dellwig et al., 2010, 2018) and LD (Glockzin et al., 2014; Dellwig et al., 2010)"**

2.  Regarding Coomassie stainable particles (CSP)
    - *Limited understanding of CSP chemical composition.*
      *CSP are a protein containing exopolymeric particles that are abundant and ubiquitous in aquatic systems. CSP are transparent, so in order to visualize them, they need first to be stain with Coomassie Brilliant Blue, a dye commonly used to stain proteins (Bradford, 1976).*
      *Phytoplankton (diatoms and cyanobacteria) and heterotrophic bacteria are sources of CSP. Scarce studies had examining CSP properties and dynamics; it has been suggested that CSP are less sticky and more labile than TEP (Thornton, 2018), but there is no information about the amount of carbon and nitrogen that CSP contain or their specific composition regarding amino acids.*
      *The limited information about the chemical composition of CSP constrains our understanding of the relationship between CSP and other biogeochemical relevant parameters such as the amino acid-based degradation index (DI). Theoretically, if CSP are enriched in labile proteins, we could expect a significant correlation with a DI that indicate fresher material (positive). Our results showed that there is not a significant relationship between TAA flux and CSP flux (r2=0.2, p=1); but there was a significant positive relationship between CSP flux and DI (R2=0.84, P<0.005, data not shown). However, due to the lack of information on their exact chemical composition, this assumption would need to be explored further before establishing a relationship.*
      *In this research, we studied CSP to investigate if they aggregate with particles similar to TEP do; our results showed that when MnOx-like particles are abundant in the water column, they aggregate not only with TEP but also with CSP (Figure 6).*
    - **ACMS:**

      **We added a paragraph in the introduction indicating the significance of CSP for this study, L155-161: "Another type of less studied exopolymer particles are Coomassie stainable particles (CSP), they are protein-containing particles that stain with Coomassie brilliant blue (Long and Azam 1996). Little is known about the characteristics and dynamics of those particles in marine systems and their potential to form aggregates with MnOx had not been studied. Different to TEP, CSP have a limited role on the aggregation of diatoms (Prieto et al., 2002; Cisternas-Novoa et al., 2015), but seem to be important for the aggregation of cyanobacteria (Cisternas-Novoa et al., 2015)"**

      **We change the starting sentence about TEP and CSP in the method section, L234-235 "Polysaccharide (TEP) and protein (CSP) exopolymer particles, from sediment**

**trap and water column samples were analyzed by microscopy according to Engel (2009)."**

**We add a sentence to the method section indicating the specific dyes that are used to studied TEP and CSP, L235-239 "Duplicate aliquots of 5 to 20 mL were filtered onto 0.4 ⬜m Nuclepore membrane filters (Whatmann) and stained with 1 mL of Alcian Blue solution, a dye that target acidic polysaccharides, for TEP or 1 mL of Coomassie brilliant blue solution, a dye commonly used to stain proteins (Bradford, 1976), for CSP."**

Minor comments:

Line 16: "Sinking particles are the main form in which photosynthetically fixed carbon is transported from the euphotic zone to the ocean interior, the so called biological pump (BCP)". And of course you can discuss the Baltic being an ocean.

*AR: We modify this sentence.*

**ACMS: L18-19, this sentence was modified to "Particle sinking is a major form to transport photosynthetically fixed carbon below the euphotic zone via the biological carbon pump (BCP)"**

Line 22: GB, but not

*AR: We added a comma to this sentence*

**ACMS: L24, this sentence was modified to "GB, but not"**

Line 25: oxygenated by the inflow of relatively saline waters from the North Sea?

*AR: We modified this paragraph and include the editor recommendation*

**ACMS: L23-25, this sentence was modified to "The two basins showed different oxygen regimes resulting from the intrusion of oxygen-rich water from the North Sea that ventilated the water column below 140 m in GB, but not in LD."**

Line 28: POC has not been defined. Abbreviations and acronyms are typically defined the first time you use them in the abstract and again in the main text. If you only use them once, just use the full name.

*AR: We defined POC in line 30, which is the first time that we used in the abstract and in text. All the abbreviations definitions were checked to make sure we defined them the first time that we used*

**ACMS: L32, this sentence was modified to: "particulate organic carbon (POC)"**

Line 48: We're in the main text now so define BCP again.
Line 49: POC is defined, great, do the same in the abstract.
**ACMS: We define the followings abbreviations:**
**L35 (Abstract): chlorophyll a (Chl a)**
**L56 (main text): biological carbon pump (BCP)**
**L64: oxygen (O2)**

Line 53-54: What do these authors mean with "higher refractory nature of sinking particles in the OMZ?

*AR: This mechanism refers to the sinking fluxes associated to lithogenic minerals and refractory terrestrial OM, currently this type of OM have been mostly study in oxic water column while its contribution to POC flux in OMZ remains largely unexplored (Keil et al., 2016; Van Mooy et al., 2002).*

**ACMS: L62-63, these sentences were modified to "the potentially high contribution of refractory terrestrial organic matter (OM) to the POC flux (Keil et al., 2016; Van Mooy et al., 2002)"**

Line 58-60: Didn't you just give that information?

*AR: This paragraph aims to point out that most of the research on POC flux in OMZ has been done in the tropical ocean; thus, there is not sufficient knowledge about how low O2 could affect POC flux in temperate-boreal continental shelf regimes such as the Baltic Sea.*

**ACMS: L67-71, the paragraph was modified to "Currently, the study of POC vertical flux in OMZ's has been mostly focused on the tropical ocean (Cavan et al., 2017; Devol and Hartnett, 2001; Engel et al., 2017; Keil et al., 2016; Van Mooy et al., 2002); whereas, how low O2 concentration would affect the composition and fate of sinking OM, and the efficiency of the BPC in oxygen-deficient zones of temperate-boreal regimes such as the Baltic deep basins had been less studied."**

Line 70: Gotland Basin (GB)

*AR: We defined the GB abbreviation here and deleted it from the L92*

**ACMS: L81, this sentence was modified to "Gotland Basin (GB)"**

Line 73: Define OM or even POM

*AR: We defined POM in L78 (previously L73)*

**ACMS: L85, this sentence was modified to "particulate organic matter (POM)"**

Line 84: "… would be necessary the bottom water…."?

*AR: We modified this paragraph for clarity*

**ACMS: L94-97, this paragraph was modified to "However, since hypoxia occurred naturally in the Baltic Sea due to physical processes, mitigating eutrophication will only decrease the spatial extent and intensity of the O2 deficiency in the deep basins."**

Line 85: Gotland Basin, used before in line 70, define GB in line 70.

*AR: We defined GB in L75 (previously L70)*

**ACMS: L97, this sentence was modified to "GB (248 m) and Landsort Deep (LD, 460 m)"**

Line 87: just Kattegat should do I think.

*AR: We deleted "Strait"*

**ACMS: L99, this sentence was modified to "limited water exchange with the North Sea through the Kattegat"**

Line 98: GB, it has been defined now.

*AR: We replaced "Gotland Basin"  by "GB" in L105*

**ACMS: L112, this sentence was modified to "the MBI had reached GB"**

Line 105: "Water column stratification…"

*AR: This sentence was deleted because the paragraph was modified for clarity.*

Line 108: Does that redoxcline still exists during/after the MB?

*AR: The 2014/2015 MBI altered the water column profiles in the GB from March to February 2015. The oxygenated waters reached GB in March 2015, and deep water anoxia started to be re-established in July 2015, a subsequent minor inflow event re-oxygenated the deep waters of GB in February 2016. We reorganize the entire paragraph to clearly separate the "regular hypoxic conditions" from the scenario generated by the MBI.*

**ACMS: L97-117, the paragraph was modified to: "GB (248 m) and Landsort Deep (LD, 460 m) are the deepest basins of the Baltic Sea. They exhibit permanent bottom-water hypoxia (Conley et al. 2002), caused by a combination of limited water exchange with the North Sea through the Kattegat, strong vertical stratification, and high production /remineralization of OM due to eutrophication (Carstensen et al., 2014b; Conley et al., 2009). A permanent transition zone of about 2 to 10 m thickness separates the oxygenated surface and the oxygen-deficient waters, with a pelagic redoxcline located approximately between 127 and 129 m in GB, and between 79 and 85 m in LD (Glockzin et al., 2014). From the1950s to 1970s, the hypoxic zones (<60 µM) in the Baltic Sea had expanded fourfold (Carstensen et al. 2014). Salt-water inflows from the North Sea are the primary mechanism renewing deep water in the central Baltic Sea (Günter et al., 2008). A Major Baltic Inflow (MBI) occurred in 2014/2015 (Mohrholz et al. 2015); this event ventilated bottom waters for five months between February and July 2015 (Holtermann et al., 2017). This MBI caused the intrusion of O2 to deep hypoxic waters, substantial temperature variability (Holtermann et al., 2017), displacement of remnant stagnant water masses by new water that changed the chemistry of the water column (Myllykangas et al., 2017), and high turbidities that may be associated with redox reactions products (Schmale et al., 2016). At the time of sampling (June 2015), the MBI had reached GB but did not affect LD, located further northwest. The oxygenated water inflow reached GB at the beginning of March and created a secondary near-bottom redoxcline (Schmale et al., 2016); the bottom water anoxia started to re-established in July 2015 (Dellwig et al., 2018). In LD, water properties did not change due to the MBI, the sulfidic layer was maintained (hydrogen sulfide, H2S concentrations of 20.7- 21.2 ⬜M), and salinity varied between 10.6 and 10.9 (Holtermann et al., 2017)."**

Line 109: this pelagic redoxcline is the original or the second redoxcline from the previous sentence?

*AR: This refers to the pelagic redoxclines in general. In hypoxic basins like the GB, there is regularly one pelagic redoxcline in the transition between the oxygenated surface and anoxic or even sulfidic*

*bottom waters. However, as a consequence of the MBI, a second pelagic redoxcline developed where there are the conditions for the same redox reactions to occur.*

**ACMS: L118-122, this sentence was modified to "Pelagic redoxclines are the suboxic transition between oxic and anoxic - even sulfidic- waters. A steep redox gradient characterizes this transition zone were electron acceptors and their reduced counterparts are vertically segregated, and biogeochemical transformations mediated by microbial processes are actively occurring (Bonaglia et al., 2016; Brettar and Rheinheimer, 1991; Neretin et al., 2003)"**

Line 114-115: "…under oxic conditions OR in the presence of nitrate they react with O2 and …" They will get oxidized, but in the latter case there is no O2?

*AR: We deleted this sentence since we modify the paragraph for clarity*

Line 127: TEP is or TEP particles are

*AR: TEP are defined as transparent exopolymeric particles. Therefore, TEP are is correct*

Line 133: by Stokes law…

*AR: We deleted "The" and the apostrophe of according to the editor suggestion*

**ACMS: L149-150, this sentence was modified to "by Stokes law"**

Line 161: special variability?

*AR: I don't understand the editor comment "special variability" in L161 or in any other place in the manuscript. Do you want me to add "special variability" in L161, I am not sure were…*

Line 173: filled up to 10 L? How much water did you have to add to get to 10 L. And why would you dilute your samples?

*AR: In general, after pooled ( ~0.6-0.8 L per tube) 12 tubes per depth together, we had to add approximately between 0.4 and 1.5 L. We standardize the final volume in order to be able to compare between depths following the procedure described in (Engel et al., 2017)*

**ACMS: L201-204, we modified the paragraph for clarity to "Then, we pooled together the remaining water, containing the sinking material (~0.6-0.8 L), of 12 tubes per depth into a large container, that we filled-up to 10 L with filtered seawater (between 0.4 and 1.5 L) to have the same volume per depth."**

Line 174: swimmers were removed with a 500 μm mesh screen?

*AR: Yes, we use a 500 μm to screen the trapped material and remove the larger swimmers; this is one of the methods commonly used for swimmer removal (Buesseler et al., 2007; Conte et al., 2001). A reference was added to line 188*

**ACMS: L188, we added a reference "…the samples were screened with a 500 ⬚m mesh to remove swimmers (Conte et al., 2001)."**

Line 185: Aliquots from the 10 L? So not only trapped material, but diluted trapped material?

*AR: We pooled together 12 tubes per depth, ~0.6-0.8 L per tube, this volume contains the trapped material already diluted in the saline solution that we added before deployment (50 g L-1 of NaOH in 0.2 μm filtered seawater). The filtered seawater added to standardize the volume to 10 L per depth after recovery did not significantly increase the dilution of the trapped material. We measured the POM concentration in the saline solution, and we use that as a blank, the POM concentration was always negligible in the blank compared to the trapped material.*

**ACMS: L201-204, we modified the paragraph for clarity to "Then, we pooled together the remaining water, containing the sinking material (~0.6-0.8 L), of 12 tubes per depth into a large container, that we filled-up to 10 L with filtered seawater (between 0.4 and 1.5 L) to have the same volume per depth"**

Line 192: Did you define POP?

*AR: Yes, it is defined in L2014 (previously L184)*

Line 200-209: Am I correct in thinking that everything between 5 and 20 μm is both counted by flow cytometer and microscopy? I am assuming the flow cytometer is counting cells containing chlorophyll and/or phycoerythrin?

*AR: Yes, that is correct, although in practice the organisms quantified by microscopy ranged between 10 and 200 μm (L204); thus everything between 10 and 20 μm was counted by flow cytometer and microscopy.*
*As the editor points out, the flow cytometer is counting cells containing chlorophyll and/or phycoerythrin (L208)*

Line 214: Explain what coomassie normally stains. Later on explain why that might be interesting.

*AR: As we mention in our reply to the general comments, CSP are a protein containing exopolymeric particles that are abundant and ubiquitous in aquatic systems. CSP are stain with Coomassie Brillant Blue, a dye commonly used to stain proteins (Bradford, 1976). We added a description of CSP in the introduction to define and explain their significance in this study. We also modified the methods section to better explain CSP.*

**ACMS: L155-161, we added a paragraph to the introduction explaining the significance of CSP in this study "Another type of less studied exopolymer particles are Coomassie stainable particles (CSP), they are protein-containing particles that stain with Coomassie brilliant blue (Long and Azam 1996). Little is known about the characteristics and dynamics of those particles in marine systems and their potential to form aggregates with MnOx had not been studied. Different to TEP, CSP have a limited role on the aggregation of diatoms (Prieto et al., 2002; Cisternas-Novoa et al., 2015), but seem to be important for the aggregation of cyanobacteria (Cisternas-Novoa et al., 2015)."**

**L234-235, we modified the sentence to "Polysaccharide (TEP) and protein (CSP) exopolymer particles, from sediment trap and water column samples were analyzed by microscopy according to Engel (2009)."**

**L231-233 "Alcian Blue solution, a dye that target acidic polysaccharides, for TEP or 1 mL of Coomassie brilliant blue solution, a dye commonly used to stain proteins (Bradford, 1976), for CSP."**

Line 231-236: If this is a known method to measure particles containing MnOx, refer to these papers or methods. MnOx-like particles could basically be anything.

*AR: This point was explained in our reply to the general comments (1. Regarding MnOx-like particles)*

**ACMS: L252-266, we added a paragraph explaining how we identified and quantified MnOx-like particles: "MnOx-containing particles have been commonly identified based on their morphology, size and elemental composition, confirmed by scanning electron microscopy (SEM) and energy dispersive x-ray microanalysis (EDX) (Neretin et al., 2003; Glockzin et al., 2014; Dellwig et al., 2010, 2018). In this study, we did not measure the elemental composition of the particles. Thus, we identified them as "MnOx-like particles" based on similar morphology, size, and association with organic matter (OM) as MnOx-containing particles previously described in the Baltic Sea (eg., Neretin et al., 2003 and Glockzin et al., 2014). The abundance and size of MnOx-like particles were determined using particle recognition on filters and imaging processing similar to the method used by Neretin et al. (2003) but without the chemical composition analysis of the particles. For the image analysis, we used the same images as for TEP and CSP analysis and modified image analysis procedure described above as follows: thirty images per filter (200x) were analyzed semi-automatically using ImageJ software. After RGB split, the blue channel pictures were used to quantify MnOx-like particles in the water column and sediment traps. In this manner, the MnOx-like particles were clearly visible with a negligible disruption from TEP or CSP stained blue."**

Line 243-244 ".... Undergoes degradation..."

*AR: We deleted "it" and "to"*

**ACMS: L274, this sentence was modified to "undergoes degradation"**

Line 272: The deepest point sampled in the LD (430 m)

*AR: We modified the sentence according to the editor's recommendation*

**ACMS: L319-320, this sentence was modified to "from 74 m to the deepest point sampled in the LD (430 m)"**

Line 299: increased to 38.9

*AR: We the word "to"*
*(The line that the editor indicates (L299) did not coincide with the line numbered (L297) in the last version online form Nov 15, 2018)*

**ACMS: L343, this sentence was modified to "increased to 38.9"**

Line 301: the lowest concentration was not at 180 m?
*(L299)*
*AR: No, H2S was not measurable at 300 and the lowest concentration was measured at 350 (0.04 $\mu$M), this coincide with a peak on NO2 (Fig2b).*

Line 304: GB µg per liter, LD µg per liter and µM?
*(L302)*
*AR: We deleted "and 0.1-0.3 µM", it was a typo*

**ACMS: L347, this sentence was modified to "in LD (1.4-1.2 µg L-1, Fig. 3e)"**

Line 304-312: There is overlap in your defined pico- and nanophytoplankton (< 20 µm) and large phytoplankton (> 5 µm)?
*(L309)*
*AR: We deleted "large" to avoid confusion due to the overlap; thus, we refer only to the different method that we use to quantify phytoplankton and not to their size.*

**ACMS: L354, this sentence was modified to "Phytoplankton (>5 µm) abundance, determined by microscopy,)"**
**L355-356: "Filamentous cyanobacteria dominated the phytoplankton community at both stations with up to 90% corresponding to Aphanizomenon sp."**

Line 314: Filament counts, so the actual biomass or even individual cell counts could be even way larger, right? This also means that the other spp are a percentage of the total **counts** or phytoplankton **counts**. Not of the total biomass or even cell counts. Completely dwarfed by the cyano's therefore, right?
*(L310)*
*AR: We agreed with the editor that the individual cell counts must be significantly larger. The other spp are a percentage of the phytoplankton counts (considering filament counts), which means a significant dominance of cyano.*

Line 315: total counts, so you could wonder if they are indeed significant based on biomass or cell counts taking into account you counted filaments rather than cyano cells.
*(L312)*
*AR: We deleted "total" to make clear that we are talking about phytoplankton counts in which we considering filament counts.*

Line 317: total phytoplankton counts
Line 318: phytoplankton counts

*AR: We are not sure if we could follow this correctly since the lines that the editor indicate did not coincide with the line numbered in the last version online form Nov 15, 2018. We deleted "total" to avoid confusion since we are always referring to the phytoplankton counts considering filament counts.*

**ACMS:**
**L356 (previous L312) "represented 56% of the phytoplankton counts in the GB"**
**L358 (previous L313) "Dinoflagellates (dominated by Dinophysis sp.) were significant in both stations (19% of the phytoplankton counts)**
**L360 (previous L315) "(25% and 4% of the phytoplankton counts respectively)"**

Line 331: TEP particles were counted right? So shouldn't this be counts as well?

*AR: As described in the methods section (L244-247) "Additionally, TEP and CSP in water samples from the stations where we deployed sediment traps were analyzed spectrophotometrically (with higher*

*vertical resolution than microscopy) according to Passow and Alldredge (1995) and Cisternas-Novoa et al. (2014), respectively"*

Line 340: so here is a little indication of what CSP might be… gel like, the question what it is still remains.

*AR: As explained above (please see replay for L214), we now described CSP, their protein-containing nature, and their significance for this study in the introduction; in addition, we add that Coomassie brilliant blue stains proteins in the methods section.*

Line 345: MnOx containing particles? But did you actually measure MnOx? Or just counted particles that looked and behave as MnOx?

*AR: As explained above (please see reply for L231-236), we now added a paragraph explaining how we identified and quantified MnOx-like particles (L252-266) to the methods section. Since we did not perform SED-EDX analysis (but counted and size particles that looked and behave as MnOx-containing particles in many ways, please see reply to general comments "1. Regarding MnOx-like particles") we call the particles described in this study "MnOx-like particles" and compared them with the previously described MnOx-containing particles.*

Line 362-363: by 18%
*(L360-361)*
*AR: We changed (18%) for by 18%*

**ACMS: L406, this sentence was modified to "POC flux slightly increased by 18% from the shallowest (40 m) to the deepest (180 m) sediment trap"**

Line 376: Fully oxygenated water depths
*(L374)*
*AR: We added "water"*

**ACMS: L418-419, this sentence was modified to "MnOx-like particles were not observed in sediment trap samples collected in fully oxygenated waters depths (40 and 60 m)"**

Line 384-387: IS the TAA related to CSP?

*AR: CSP are protein-containing particles, but there is no information about their specific chemical composition, this limits our understanding of the relationship between CSP and amino acids. Considering our data there is not a significant relationship between TAA flux and CSP flux ($r2- =0.2$, $p=1$, not include in the ms).*
*On the other hand, since one of the sources of CSP are phytoplankton exudates, it could be expected that CSP were related to amino acids released by phytoplankton (Thornton et al., 2018). Diatoms commonly released serine, glycine, glutamic acid, aspartic acid, ornithine, and histidine (Myklestad 2000); phytoplankton also releases fluorescent amino acids such as tryptophan, tyrosine, and phenylalanine.*
*In our data set, there is a significant relationship between CSP flux and the concentration of glutamic acid ($r2=0.73$, $p<0.01$), serine (($r2=0.70$, $p<0.01$), alanine ($r2=0.77$, $p<0.01$), aspartic acid ($r2=0.65$, $p<0.05$), tryptophan ($r2=0.64$, $p<0.05$), tyrosine ($r2=0.83$, $p<0.01$), and phenilalanine ($r2=0.77$, $p<0.01$). There was not a significant relationship with glycine, and we did not measure serine, histidine or ornithine.*

*However, since there is no studies about the actual amino acids content of CSP, those relationships may be interpreted with caution and further research focusing on the composition of CSP are needed to conclude in this respect.*

Line 398: SOM or DOM?
*AR: Aminoacids and sugars were measured in total organic matter (DOM +POM), we compared the ratios from the material collected in the water column, called suspended organic matter, vs the material collected in sediment traps, called sinking organic matter.*

**ACMS: L436-437, the opening sentence of this paragraph was modified for clarity "Comparing molar elemental ratios of sinking (from sediment trap material) and suspended (from water column) particles to the revisited Redfield ratio"**

Line 406: indicate the Redfield ratio for Si.

*AR: We add the Si to the modified Redfield ratio*

**ACMS: L437-438, the Redfield ratio was modified to "to the revisited Redfield ratio for living plankton (106C: 16N: 15Si: P; Redfield et al., 1963; Brzezinski, 1985)"**

Line 411: TCHO?
(L409)
*AR: We added the "T"*

**ACMS: L458-459, this sentence was modified to "Similarly, the carbon contained in TCHO made up a larger percentage in sinking than in suspended particles (Table 5)"**

Line 427: biomass? Or counts?
(L425)
*AR: We changes biomass for abundance (estimated by counting)*
**ACMS: L474-477, this sentence was modified to "Moreover, though there were slight differences between the stations concerning phytoplankton abundance and composition, and concentration and chemical composition of POM, in the surface water column, those were not significant."**

Line 437: the overlap in size classes again?
(L435)
*AR: We changed "smaller" by "measured by flow cytometer" (L431) and we deleted "larger" (L435) to differentiate the phytoplankton quantified by different methods and not by size (consistently with the methods section)*

**ACMS: These sentence were modified to:**
**L480-482 (previously L455): "At both stations, the abundance of pico-phytoplankton (<2 ⬜m) was an order of magnitude higher than nano-plankton (Table 2)."**
**L459 (previously L435): "Microscopic analysis of phytoplankton"**

Line 483: MnOx-like and in line 485 you are sure it is actually MnOx, what has changed?
(L481, L483)
*AR: In L481 we are referring to the particles measured in this study, "MnOx-like particles" while in L483 we refer to previous findings. We added the word "containing" and the citation for clarity.*

**ACMS: L544-545 (previously L483-482), these sentences were modified to "One consequence of those changes is the vertical extension of the layer in which MnOx-containing aggregates could form (Schmale et al., 2016);"**

Line 531-532: So in Glockzin et al they actually measure manganese?
(L529-L530)
*AR: Yes, they measured particulate Mn in the water column and the Mn content in MnOx containing particles using SEM-EDX (Please see reply to general comments "1. Regarding MnOx like particles" for more details).*

**ACMS: we added a paragraph in the discussion explaining why our MnOx-like particles may correspond to the previously reported MnOx-containing particles:**

**L611-617: "The high flux of POC at GB coincided with the appearance of dark, star-shaped particles that we defined as MnOx-like particles, particularly evident at GB (Fig. 6a,b, and e), but also present in LD. Based on their morphology, size, and aggregation with OM, we propose that those particles correspond to MnOx-containing particles enriched in OM that have been previously described at GB (Neretin et al., 2003; Pohl et al., 2004; Glockzin et al., 2014; Dellwig et al., 2010, 2018) and LD (Glockzin et al., 2014; Dellwig et al., 2010)."**

Line 533: ($H_2S$), pretty sure there is $H_2O$.
(L531)
*AR: We changed H2O to H2S*
**ACMS: L641-642, the sentence was modified to: "Below the oxycline, and due to the presence of H2S, the particulate Mn concentration decreased drastically."**

Line 537-538: "… redox conditions favorable for the formation of MnOx resulting in the high MnOx flux measured …"
(L535-L536)
*AR: We changed this sentence according to the editor recommendation*

**ACMS: L646-648, these sentences were modified to: "…redox conditions favorable for the formation of MnOx, resulting in the high MnOx-like particles flux measured in the sediment trap located in the core of the OMZ (110 m) and at 180 m (oxygenated deep water)"**

Line 547: containing or should it be like?
(L545)
*AR: We changed "MnOx-containing" to "MnOx-like" since we refer to the particles measured in this study*

**ACMS: L656-657, the sentence was modified to: "The presence of MnOx-like particles in aggregates (Fig 6a) may have implications for the vertical flux of POC, PN and POP in a stratified system with a pelagic redoxcline like the Baltic Sea."**

Line 590: we consider
(L588)
*AR: We changed "considered" to "consider"*

**ACMS: L698, the sentence was modified to: "If we consider a mixed aggregate …"**

Line 592: mixed (L590)
*AR: We changed "mix" to "mixed"*

**ACMS: L701, the sentence was modified to: "…the largest mixed aggregates composed of MnOx and TEP"**

Line 598: mixed
(L596)
*AR: We changed "mix" to "mixed"*

**ACMS: L706-707, the sentence was modified to: "There is no information about the amount of OM relatively to MnOx-containing particles in those mixed aggregates"**

Line 606: than rather than that?
(L604)
*AR: We changed "that" to "than"*

**ACMS: L632, the sentence was modified to: "OM export is different under anoxic than under oxic conditions in the Baltic Sea"**

Line 615: "… how similar the biogeochemical conditions were …"
(L613)
*AR: We changed the sentence for clarity and accuracy*

**ACMS: L734-735, the sentence was modified to: "In the sections above we compared the biogeochemical conditions and the size of the POM pool in the euphotic zone of GB and LD."**

Line 625: Hence the $N_2$ fixing cyano's?
(L623)
*AR: We deleted this sentence, since it refer to the results of that particular study and the role of N2 fixation in the C:N ratio is outside the scope of this manuscript.*

**ACMS: L741-743, the sentence was modified to: "Our measured values of POC:PN (~10) and POC:POP (between 89 and 506) in suspended OM coincide with the simulated ratio reported immediately after the culmination of the spring bloom by Kreus et al. (2015)."**

Line 660: transporting solid material from the and ?
(L658)
*AR: We fixed this sentence*

**ACMS: L796-798, the sentence was modified to: "The 2014/2015 MBI supplied oxygen-rich waters to GB transporting solid material from shallower areas and modifying the O2 vertical profile and the redox conditions in the otherwise permanent suboxic deep waters."**

Line 662: for the comparison of POM

(L660)
*AR: We changed the sentence according to the editor recommendation*

**ACMS: L798-800, the sentence was modified to: "This event did not affect LD allowing for the comparison of POM fluxes and composition under two different O2 concentrations with similar surface water conditions."**

[revised manuscript text omitted]

Fig. 1

[Figure]

Fig. 2

[Figure]

Fig. 3

[Figure]

Fig. 4

[Figure]

Fig. 5

[Figure]

Fig. 6

[Figure]

Fig. 7

---

## Author Response (AR3)

Associate Editor Decision: Publish subject to technical corrections (06 Feb 2019) by Marcel van der Meer

Comments to the Author:

Dear Carolina Cisternas-Novoa and co-authors,

Thank you very much for the extensive amount of work you have put into answering my questions and improving your manuscript. I think it has improved significantly and I am very happy to accept this manuscript for publication.

Best regards,

Marcel

*Dear Marcel van der Meer,*

*We want to reiterate our gratitude for the time and effort that you and the reviewers dedicate to improve our manuscript, we appreciate it.*

*Best regards,*

*Carolina*

Non-public comments to the Author:

Dear Carolina,

I have few very minor comments based on the annotated version of your manuscript.

*Reply to the Associate Editor non-public comments,*

*Dear Marcel,*

*Following you will find the detailed changes that we did to the manuscript according to your latest minor comments. The manuscript was updated accordingly.*

*Best regards,*

*Carolina*

Line 18, abstract: Particle sinking is a major form of transport for photosynthetically fixed carbon to below the ….

*AR: We added the word "to" to this sentence*

**ACMS: L18, this sentence was modified to "Particle sinking is a major form to transport photosynthetically fixed carbon to below the euphotic zone via the biological carbon pump (BCP)"**

Line 25: , but not in the LD druing the time of sampling.

*AR: We added the "during the time of sampling" to this sentence*

**ACMS: L25, this sentence was modified to "but not in LD during the time of sampling"**

Line 49: aggregates

*AR: We replace "aggregate" by "aggregates" in this sentence*

**ACMS: L49, this sentence was modified to "...that POM aggregates with MnOx-like particles…"**

Line 70: BCP

*AR: We corrected the typo in this sentence*

**ACMS: L70, this sentence was modified to "...the efficiency of the BCP in oxygen-deficient zones …"**

Line 71: has been less

*AR: We replace "had" by "has" in this sentence*

**ACMS: L71, this sentence was modified to "...has been less studied."**

Line 93: reduce the OM

*AR: We replace "in" by "the" in this sentence*

**ACMS: L93, this sentence was modified to "...may reduce the OM downward flux..."**

Line 127: dominate

*AR: We replaced "dominates" by "dominate" in this sentence*

**ACMS: L127, this sentence was modified to "Under anoxic conditions dissolved reduced Mn forms dominate..."**

Line 129: 3 µM is reported for the LD (Dellwig et al., 2012). Van Hulten et al. (2017)….

*AR: We modify the reference to "Van Hulten et al. (2017)"*

**ACMS: L129, this sentence was modified to "Van Hulten et al. (2017) estimated an aggregation threshold for manganese oxides of 25 pM"**

Line 441: with depth

*AR: We replaced "deep" by "depth" in this sentence*

**ACMS: L441, this sentence was modified to "…while in LD it varied between 11.1 and 15.4 without a clear trend with depth"**

Line 488: …predominant phytoplankton type…

*AR: We added the word "phytoplankton" to this sentence*

**ACMS: L488, this sentence was modified to "…were numerically the predominant phytoplankton type"**

Line 491: coincided

*AR: We replaced "coincide" by "coincided" in this sentence*
**ACMS: L491, this sentence was modified to "…this coincided with a small peak in BSi concentration…"**

Line 492: diatom proportion

*AR: We replaced "diatoms" by "diatom" in this sentence*

**ACMS: L492, this sentence was modified to "…the diatom proportion …"**

Line 513: precedes

*AR: We replaced "preceded" by "precedes" in this sentence*

**ACMS: L491, this sentence was modified to "…that precedes the cyanobacteria summer bloom,"**

Line 515: constant increase?

*AR: We replaced "increment" by "increase" and modified this sentence*

**ACMS: L515, this sentence was modified to "…in GB constantly increase from mid-May to mid-June 2015…"**

Line 520: upon phosphate

*AR: We deleted the word "the" in this sentence*

**ACMS: L520, this sentence was modified to "…the termination of the summer bloom depends upon phosphate availability …"**

Line 522: after the summer bloom,

*AR: We added the word "the" in this sentence*

**ACMS: L520, this sentence was modified to "…after the summer bloom …"**

Line 596: relatively high or higher

*AR: We modified this sentence according to the editor suggestion*

**ACMS: L596, this sentence was modified to "…is relatively high in the Baltic Sea compared with late fall and winter."**

Line 619: changes?

*AR: We replaced "change" by "changes" in this sentence*

**ACMS: L619, this sentence was modified to "…probably due to the oxygenation and changes in the deep water redox conditions …"**

Line 628: coincided

*AR: We replaced "coincide" by "coincided" in this sentence*

**ACMS: L628, this sentence was modified to "…which coincided with the high flux of MnOx-like particles."**

And perhaps you could acknowledge the reviewers for their help in making this the manuscript it is.

*AR: We agreed with the editor, we modified the acknowledge section accordantly*

**ACMS: L838-840, The following paragraph was added to the acknowledgements section:**

[revised manuscript text omitted]

Fig. 1

[Figure]

Fig. 2

[Figure]

Fig. 3

[Figure]

Fig. 4

[Figure]

Fig. 5

[Figure]

Fig. 6

[Figure]

Fig. 7